# Daily electric field treatment improves functional outcomes after thoracic contusion spinal cord injury in rats

Bruce Harland [1,4] ✉, Lukas Matter [2,4] ✉, Salvador Lopez [1], Barbara Fackelmeier[3], Brittany Hazelgrove[1], Svenja Meissner[1], Simon O'Carroll [3], Brad Raos [1], Maria Asplund [2,5] & Darren Svirskis [1,5]

Spinal cord injury (SCI) can cause permanent loss of sensory, motor, and autonomic functions, with limited therapeutic options available. Low-frequency electric fields with changing polarity have shown promise in promoting axon regeneration and improving outcomes. However, the metal electrodes used previously were prone to corrosion, and their epidural placement limited the penetration of the electric field into the spinal cord. Here, we demonstrate that a thin-film implant with supercapacitive electrodes placed under the dura mater can safely and effectively deliver electric field treatment in rats with thoracic SCI. Subdural stimulation enhanced hind limb function and touch sensitivity compared to controls, without inducing a neuroinflammatory response in the spinal cord. While axon density around the lesion site remained unchanged after 12 weeks, in vivo monitoring and electrochemical testing of electrodes indicated that treatment was administered throughout the study. These results highlight the promise of electric field treatment as a viable therapeutic strategy for achieving long-term functional recovery in SCI.

Spinal cord injury (SCI) disrupts the communication between the brain and body, with severity varying based on the level and extent of damage. Symptoms include motor and sensory loss, neuropathic pain, and bladder, bowel, and sexual dysfunction. No cure for SCI exists, however, even minor improvements in function would enhance patient's quality of life[1]. Electrical stimulation as a treatment modality has been translated from basic science to clinical applications[2,3]. Common modalities for electrical stimulation after SCI are: high-frequency neuromodulation (μs pulses) to bypass the injury[2,4,5], and low-frequency electric field (EF) treatment (ms−min pulse widths) to promote axonal regeneration and reconnection across the lesion[3,6,7]. Notably, these technologies are compatible and complementary, as neuromodulation could provide targeted restoration of motor

function while at the same time EF treatment could provide regeneration of injured tracts to form long-lasting neuronal connections. However, despite promising evidence of regeneration and improved outcomes after SCI, EF treatment has encountered challenges due to the increased demands that low-frequency stimulation places on the electrode materials.

EF treatment has its origin in early in vitro studies showing that direct current stimulation could direct and promote axonal regeneration towards the cathode[8,9]. Using sets of electrodes sutured to muscle above the dura mater, this approach was then demonstrated to regenerate axons, to reduce axonal degeneration, and to recover muscle reflexes in transected spinal cords of lamprey fish and Guinea pigs[7,10]. In a contusion injury model in rats, similar direct current

[1]School of Pharmacy, University of Auckland, Auckland, New Zealand. [2]Department of Microtechnology and Nanoscience, Chalmers University of Technology, Gothenburg, Sweden. [3]Department of Anatomy and Medical Imaging, School of Medical Sciences, University of Auckland, Auckland, New Zealand. [4]These authors contributed equally: Bruce Harland, Lukas Matter. [5]These authors jointly supervised this work: Maria Asplund, Darren Svirskis. ✉e-mail: bruce.harland@auckland.ac.nz; lukas.matter@chalmers.se

stimulation resulted in higher amplitude of motor-evoked potentials, improved performance on an inclined-plane test, and greater cell counts in motor-related brain regions compared with non-treated controls[11,12]. A refinement was then made to alternate the polarity of stimulation every 15 mins (-0.5 mHz), which promoted axon outgrowth in both directions while preventing die-back from the anode[13]. This treatment was deployed in patient dogs implanted with epidural stimulators resulting in significant improvements in superficial and deep pain response below the lesion compared with controls, whereas neurologic scores for ambulation and proprioceptive placing were not significant[6,14]. Similarly, human SCI patients treated via implanted stimulators had significantly improved light touch and pinprick sensation but in addition also had higher motor scores after EF treatment as well as recovery or improvement of somatosensory-evoked potentials[3]. However, the metallic electrodes available for these prior studies limited progression of the low-frequency EF treatment paradigm with corrosion hampering their effectiveness and potentially generating toxic concentrations of byproducts such as metal ions, pH changes and reactive oxygen species damaging local tissues[14–16].

Our work heralds a return to this promising approach by leveraging thin-film fabrication, state-of-the-art electrode materials, and subdural positioning to enhance the power, precision and biocompatibility of stimulation. We have previously developed an ultra-thin spinal implant for recording that can be safely inserted beneath the dura mater to make direct contact with the pial surface of the spinal cord with no permanent impairment of hind limb function or other welfare issues in non-injured rats[17]. Although requiring a more invasive surgery, a subdural implant allows for stimulation that does not need to penetrate the dura mater and arachnoid membranes as well as the subarachnoid space filled with motional and highly conductive cerebrospinal fluid[18]. We expect this to allow subdural electrodes to deliver a stronger stimulation while using an order of magnitude less power input and importantly much greater penetration of the EF within the spinal cord[19,20]. The lower current requirements reduce the propensity for irreversible faradaic reactions at the electrode-tissue interface. We have further enhanced the biotolerability of the stimulation using pseudocapacitive materials as electrodes. We have augmented the implant with larger 200 μm diameter dedicated stimulation electrodes enhanced with sputtered iridium oxide films (SIROF). SIROF electrodes are capable of pseudocapacitive charge injection through a mix of reactions where iridium transitions through different valence states, which generate surface-bound reaction products that do not diffuse into local tissue and thereby have a high degree of reversibility when stimulation is charge balanced[21–23]. However, even these revisions would not allow for 15-min pulses of stimulation to be delivered within the safe window of applied charge density. Therefore, we have adopted the use of 250 ms pulse width stimulation (2 Hz), which is still 1000 times longer pulses than the typical neuromodulation. The 2 Hz stimulation has previously been demonstrated to be effective for directing and encouraging axonal growth in vitro. In that study, 3D-cultured rat cortical neurons were exposed to alternating fields ranging from 0.5 Hz to 2 kHz for up to 4 days, and the greatest effect of ~30% axon length increase was achieved with stimulation of 2 Hz[24].

EF treatment is one of the few therapies shown to regenerate axons tracts and improve functional outcomes after SCI, however, it has to date not been able to be administered safely and effectively due to the demands placed on metal electrodes by low-frequency stimulation and their epidural implantation. Here, we show how this is made possible using an ultrathin subdural implant with SIROF electrodes to administer a daily 2 Hz EF treatment in rats with a thoracic contusion injury. Histological analysis showed that the EF treatment was well tolerated and did not trigger a neuroinflammatory response. We did not observe significant differences in markers related to axon density and axonal regeneration around the injury site. However, rats receiving the treatment showed significant improvement in recovery of hind limb function from week 4 onwards compared with non-treated controls and touch sensitivity significantly improved after one week of treatment and remained consistent throughout the study period. Challenges in removing implants intact after the 12-week treatment period, and an increasing in vivo impedance over the course of the experiment, added complexity to the evaluation of the consistency of the EF treatment. However, electrochemical benchtop testing and high-resolution imaging of electrode surfaces after the EF treatment verified the ability of the electrodes to successfully deliver the EF treatment while computational modeling estimated the EF strength within the spinal cord in order to validate that the stimulation met our target parameters.

## Results

### EF treatment improved the recovery of hind limb motor function and touch sensitivity in rats

We tested the effectiveness of a 2 Hz biphasic EF treatment (amplitude 5 μA, pulse width 250 ms) by surgically inserting the implant in three groups of rats (Fig. 1a, b). Two of the groups, treated ($n = 8$) and non-treated ($n = 10$) received a 175 kilodynes impact injury at the boundary between spinal segments L1/L2 (directly below the T11 spinal process). The maximum force (Fig. 1c) of the impactions was the same (unpaired $t$-test, $p = 0.94$) and the force vs time profiles were consistent (Supplementary Fig. 1a–c) between the two groups. Beginning the day after surgery, the treated group received 1-h of EF treatment daily for 7–11 days, then on weekdays only (5 days/week) for 12 weeks. Stimulation was delivered via 200 μm ∅ SIROF electrodes positioned on either side of the injury (Fig. 1d; Supplementary Fig. 2). The implant also included two reserve electrodes as well as smaller recording electrodes, which were not part of this study (Fig. 1a).

Hind-limb function was assessed at post-surgery days 1, 3, 7, and then weekly, using the Basso Beattie Bresnahan (BBB) scale (0 = no function, 21 = normal function)[25] comparing treated rat's scores with non-treated and a no-injury control group ($n = 10$). During the first week after surgery, the treated group showed slower recovery compared to the non-treated group, with a significant difference observed on day 7 (mixed model ANOVA, Tukey's post-hoc, $p < 0.05$; Fig. 1e). However, from week 4 onwards, treated rats demonstrated significantly higher scores, indicating greater recovery of hind limb function compared to the non-treated controls (mixed model ANOVA, Tukey's post-hoc's, $p$'s < 0.05). Additionally, treated rats continued to show improvement, with significantly higher scores from week 8–12 compared to their week 5 scores (mixed model ANOVA, within-group Tukey's post-hoc's, $p$'s < 0.05). In contrast, non-treated SCI rats plateaued at week 5 and showed no further improvement (mixed model ANOVA, within-group Tukey's post-hoc's, $p$'s > 0.24), which aligns with previous work in our lab using a similar injury model[26]. By week 12, 100% of the treated rats scored 14 or higher, indicating consistent coordination between the front and hind limbs, compared to only 20% of the non-treated rats, who struggled with smooth motion due to missteps, hind paw lagging, or hopping (see Supplementary Video 1). All animals in the no-injury group showed normal locomotion over the period of the experiment (BBB Score = 21).

We assessed the recovery of mechanical sensitivity using an electronic von Frey task, in which a fine metal filament applied a ramping force (2.5 g/s) to the plantar surface of the hind paws. Treated rats withdrew their paws quicker on week 1 and 3–12 compared to the non-treated group indicating a recovery of touch sensitivity (mixed model ANOVA, Tukey's post-hoc's, $p$'s < 0.05; Fig. 1f). Furthermore, treated rats had faster withdrawal rates than the non-injury controls on week's 4 and 9 (mixed model ANOVA, Tukey's post-hoc's, $p$'s < 0.05). This difference may be attributed to the high variability of this task when administered to uninjured rats[27], but it could also suggest a degree of hypersensitivity in the paws of treated animals.

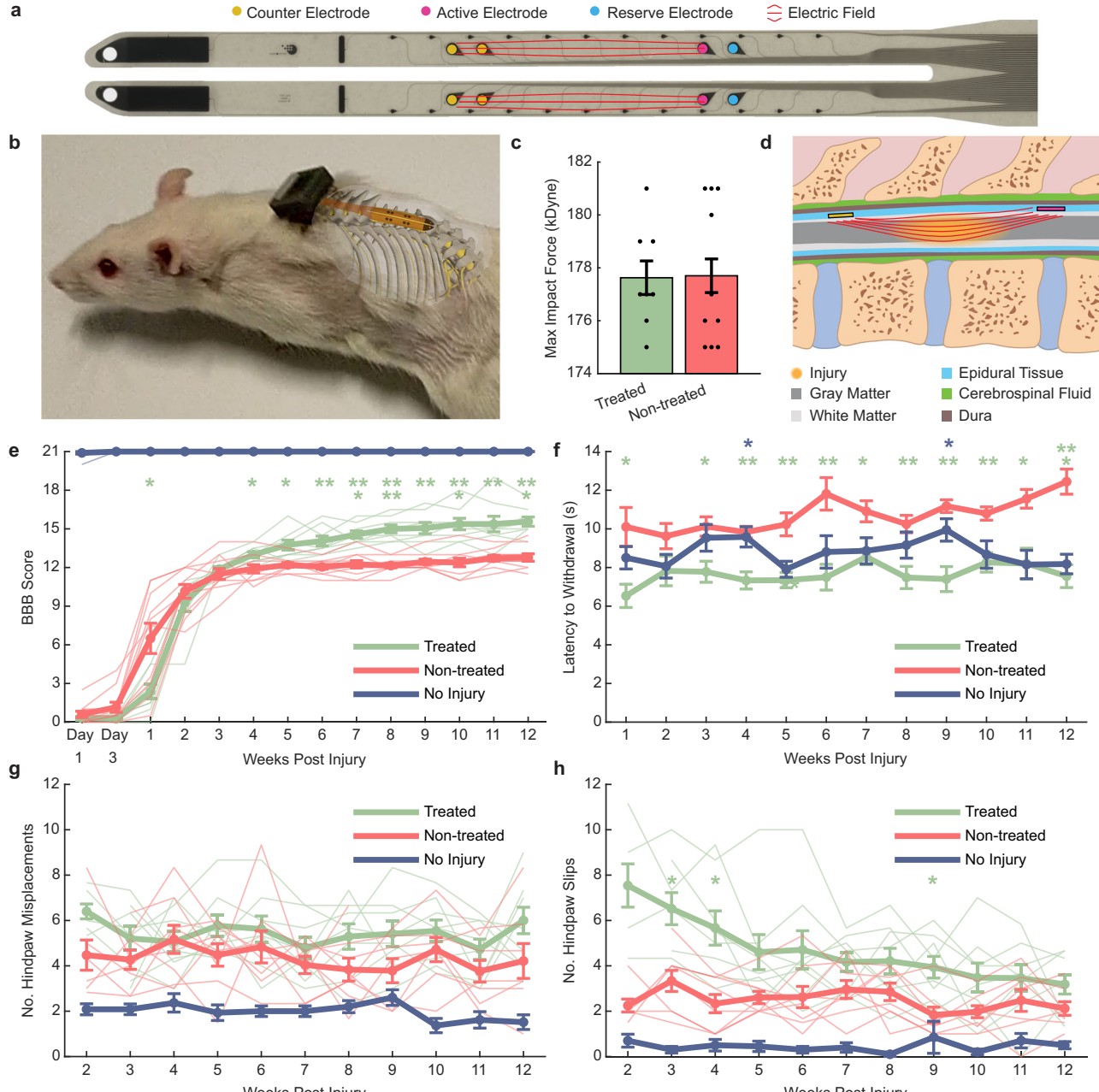

**Fig. 1 | Daily EF treatment improves hind limb motor function and touch sensitivity after SCI. a** Bioelectronic implant with stimulation electrodes. **b** Implant was inserted below T10–T12 in treated, non-treated, and no-injury rats. **c** A 175 kilodyne spinal cord impact at T11 showed no difference between groups. **d** Starting the day after surgery, treated rats received 2 Hz biphasic EF stimulation (1 h/day) via active and counter electrodes flanking the injury. **e** Treated rats initially recovered slower but surpassed non-treated rats in open-field motor scores from week 4 onward (green asterisks). **f** Treated rats showed faster hind paw withdrawal to ramping filament force at week 1 and weeks 3–12 compared to non-treated rats (green asterisks) and at week 4 and 9 compared to no-injury rats (blue asterisks). **g** Treated and non-treated rats showed similar foot misplacements on a 1-m ladder. **h** Treated rats had more slips on the ladder at weeks 3, 4, and 9 compared to non-treated (green asterisks). *$P < 0.05$, **$P < 0.01$, ***$P < 0.001$, ****$P < 0.0001$. Group

means are shown by thick lines ± SEM; thin lines represent individual animals from treated and non-treated groups in (**e, g, h**). **c, e** Treated $n = 8$, Non-treated $n = 10$, No injury $n = 10$; **f–h** Treated $n = 8$, Non-treated $n = 8$, No injury $n = 10$. **c** Two-way unpaired $t$-test, $p = 0.94$. **e–h** Two-way repeated measures ANOVA - Tukey's post hoc; **e** post hoc treated vs non-treated effects: w1: $p = 0.026$, w4: $p = 0.039$, w5: $p = 0.014$, w6: $p = 0.003$, w7: $p = 0.0002$, w8: $p < 0.0001$, w9: $p = 0.0013$, w10: $p = 0.0007$, w11: $p = 0.010$, w12: $p = 0.0002$; **f** post hoc treated vs non-treated effects: w1: $p = 0.047$, w3: $p = 0.027$, w4: $p = 0.0023$, w5: $p = 0.0066$, w6: $p = 0.0061$, w7: p = 0.016, w8: $p = 0.0092$, w9: $p = 0.0016$, w10: $p = 0.0063$, w11: $p = 0.017$, w12: $p = 0.0008$, post hoc treated vs no injury effects: w4: $p = 0.02$, w9: $p = 0.037$; **g** post hoc treated vs non-treated effects: $p$'s > 0.17; **h** post hoc treated vs non-treated effects: w3: $p = 0.011$, w4: $p = 0.011$, w9: $p = 0.017$.

Animals also performed weekly sessions in which they traversed a 1-m horizontal rung ladder, and the number of hind paw misplacements and slips were scored. This task was only attempted once animals were weight bearing on both hind paws (week 2 or 3). There was no difference in the number of foot misplacements between treated and non-treated control rats (mixed model ANOVA, Tukey's post-hoc's, $p$'s > 0.17; Fig. 1g). Treated rats had a higher number of slips compared to the non-treated rats at week 3, 4, and 9 (mixed model ANOVA, Tukey's post-hoc's, $p$'s < 0.05; Fig. 1h).

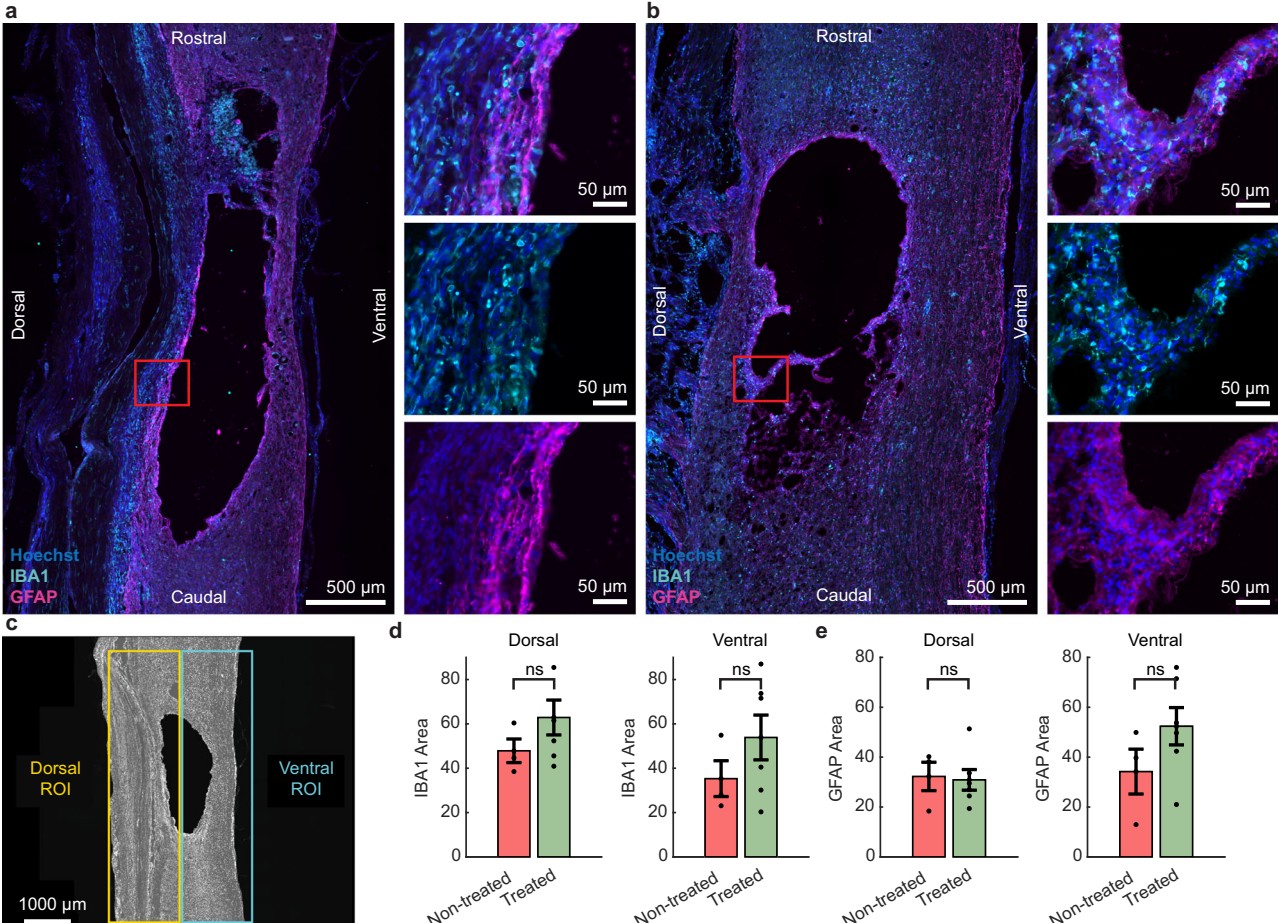

**Fig. 2 | Subdural administration of daily EF treatment did not result in a neuroinflammatory response.** Representative sagittal images are shown of immuno-labelling of IBA1 and GFAP for **a** non-treated and **b** treated rats. **c** Dorsal and Ventral regions of tissue were analyzed. **d** There was no significant difference in microglial/macrophage activity between the treated and non-treated groups in the dorsal and ventral regions of the spinal cord. **e** No difference in astrocyte activity was detected between groups. Data are presented as mean values ± SEM. **d**, **e** treated $n = 6$, non-treated $n = 3$; two-way unpaired $t$-tests. **d** Dorsal: $p = 0.30$, ventral: $p = 0.32$; **e** dorsal: $p = 0.87$, ventral: $p = 0.24$.

## Histological changes around the spinal cord injury were examined after 12 weeks

At the end of 12 weeks, rats were perfused and sagittal longitudinal sections around the injury were analyzed in both treated and non-treated groups. The size and extent of the lesion were consistent with previous studies using similar injury models[26,28] (Supplementary Fig. 11). In a previous work in non-injured rats, our findings suggest that the implant does not elicit a significant immune response[17]. In the present study, we first examined whether subdural EF treatment resulted in neuroinflammatory changes beneath the electrodes by examining markers related to microglia and astrocytes (Fig. 2a, b). Epidural EF treatment has previously been linked to increases[15] and decreases[29] in neuroinflammation, whereas a subdural treatment had not yet been evaluated. Since the stimulation current weakens as it flows ventrally due to tissue resistance, the generated EF strength will be higher in dorsal than ventral regions. If EF treatment influences neuroinflammation, larger effects would be expected in dorsal than in ventral regions. Therefore, the dorsal and ventral ROIs shown in Fig. 2c were analyzed across all histological assessments in this study (each 7.1 mm²). We did not detect significant differences in microglia/macrophage activity between the groups with comparable levels of Iba1 expression observed (unpaired $t$-test, $p$'s > 0.29; Fig. 2d). Additionally, there was no evidence of astrogliosis in response to treatment, as glial fibrillary acidic protein (GFAP) levels remained similar between groups (unpaired $t$-test, $p$'s > 0.23; Fig. 2e).

Immunohistochemistry for β-tubulin III (β-tub) and growth associated protein 43 (GAP43) was performed to assess axon density and signs of recent axonal regeneration, respectively (Fig. 3a, b). Serotonergic expression was also examined around the lesion site (Fig. 3c, d). After 12 weeks, there was no observable change in β-tub expression (unpaired t-test, $p$'s > 0.67; Fig. 3e) or GAP43 expression (unpaired t-test, $p$'s > 0.71; Fig. 3f) in injured tissue between groups. No significant difference in serotonergic expression was seen in treated animals (unpaired $t$-test, $p$'s > 0.18; Fig. 3g).

### Estimation of EF strength within the spinal cord

We verified the effectiveness of the stimulation current in generating a therapeutic EF strength by generating a finite element model of the rat's spine, guided by prior literature (Fig. 4a, see conductivities in Table 1). The model was used to estimate EF strength, with variations in white matter size and the volume of the subarachnoid space containing cerebrospinal fluid (CSF) to assess the impact of cord size. Previous studies indicated significant variation in white matter volume at L1, while gray matter remained consistent[30]. Though no intra-animal data on the size variation of the subarachnoid space was available, we hypothesized that changes in CSF volume, due to its high conductivity, would also affect EF strength. Additionally, we utilized the model to evaluate the generated EF strength of the implant placed subdurally versus epidurally.

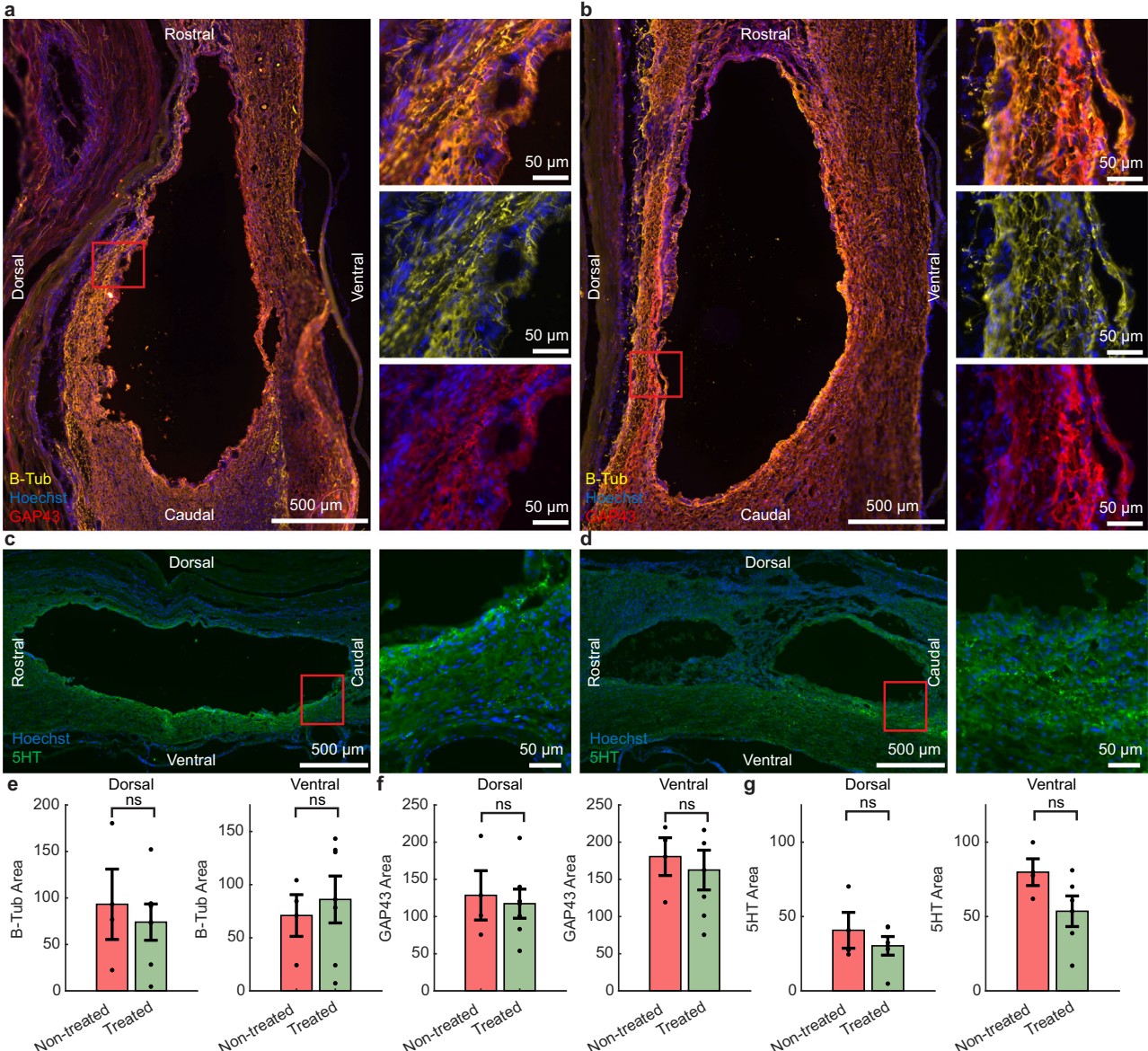

**Fig. 3 | There were no changes in axon density or serotonergic expression in rats given daily EF treatment compared with non-treated controls.**
**a** Representative sagittal images from the epicenter of the lesion with β-tub and GAP43 immunolabelling from **a** non-treated and **b** treated rats. Representative sagittal images from the epicenter of the lesion with 5HT immunolabelling from **c** non-treated and **d** treated rats. Similar expression of both **e** β-tub (overall axon density) and **f** GAP43 (axonal regeneration) were observed in both groups, either in the dorsal or ventral regions of the spinal cord. **g** Similar levels of serotonergic (5HT) expression were seen in both treated and non-treated rats. Data are presented as mean values ± SEM. **e**, **f** Treated $n = 6$, Non-treated $n = 3$; **g** Treated $n = 5$, Non-treated $n = 3$; **e**–**g** two-way unpaired $t$-tests. **e** Dorsal: $p = 0.67$, ventral: $p = 0.71$; **f** dorsal: $p = 0.79$, ventral: $p = 0.71$; **g** dorsal: $p = 0.49$, ventral: $p = 0.18$.

We simulated three cord sizes: small, medium, and large (dimensions in Table 2). Regardless of cord size, the longitudinal EF on the medial yz-plane showed higher magnitude near the electrodes and penetrated deeper ventrally, with uniform distribution between the electrodes (Fig. 4b, Supplementary Fig. 3). Longitudinal field distribution along three evaluation lines: the midline (ML), 1 mm dorsal (DL), and 1 mm ventral (VL), was similar across different cord sizes but inversely proportional in magnitude to the cord size (Fig. 4c). For the DL, the EF peaked at 4.8 mV/mm for the small cord, dropping to a minimum between the electrodes. Along the ML, the EF plateaued between 0.8 and 1.6 mV/mm across cord sizes over a 4 mm distance, while for the VL, the EF was weaker. On average, the EF in the large cord was 60% of that in the small cord, due to greater tissue volume reducing current flow. Epidural placement of the implant, rather than subdural, reduced EF strength by an average of 13% in the medium-sized spinal cord model.

## Long-term stability of SIROF electrodes

To complement the in vivo data, we investigated the stability of SIROF electrodes in 1× phosphate-buffered saline (PBS), which mimics the conductivity and composition of CSF[31]. To verify the appropriateness of our electrode materials for the selected stimulation parameters and long treatment periods, we compared the performance of SIROF-coated electrodes against Pt electrode controls. Pt electrodes have very limited pseudocapacitive charge injection capacity compared to SIROF[23], and coating Pt with SIROF greatly improved electrode stability. After 90 h of continuous stimulation in PBS with the same pulsing parameters used in vivo (Fig. 5a), the Pt stimulation electrode and neighboring stimulation and recording electrodes delaminated and dissolved, whereas SIROF electrodes remained intact (Fig. 5b, Supplementary Fig. 4a), as confirmed by high-resolution imaging, cyclic voltammetry (CV), and electrochemical impedance spectroscopy (EIS). Voltage transients during stimulation remained within the water

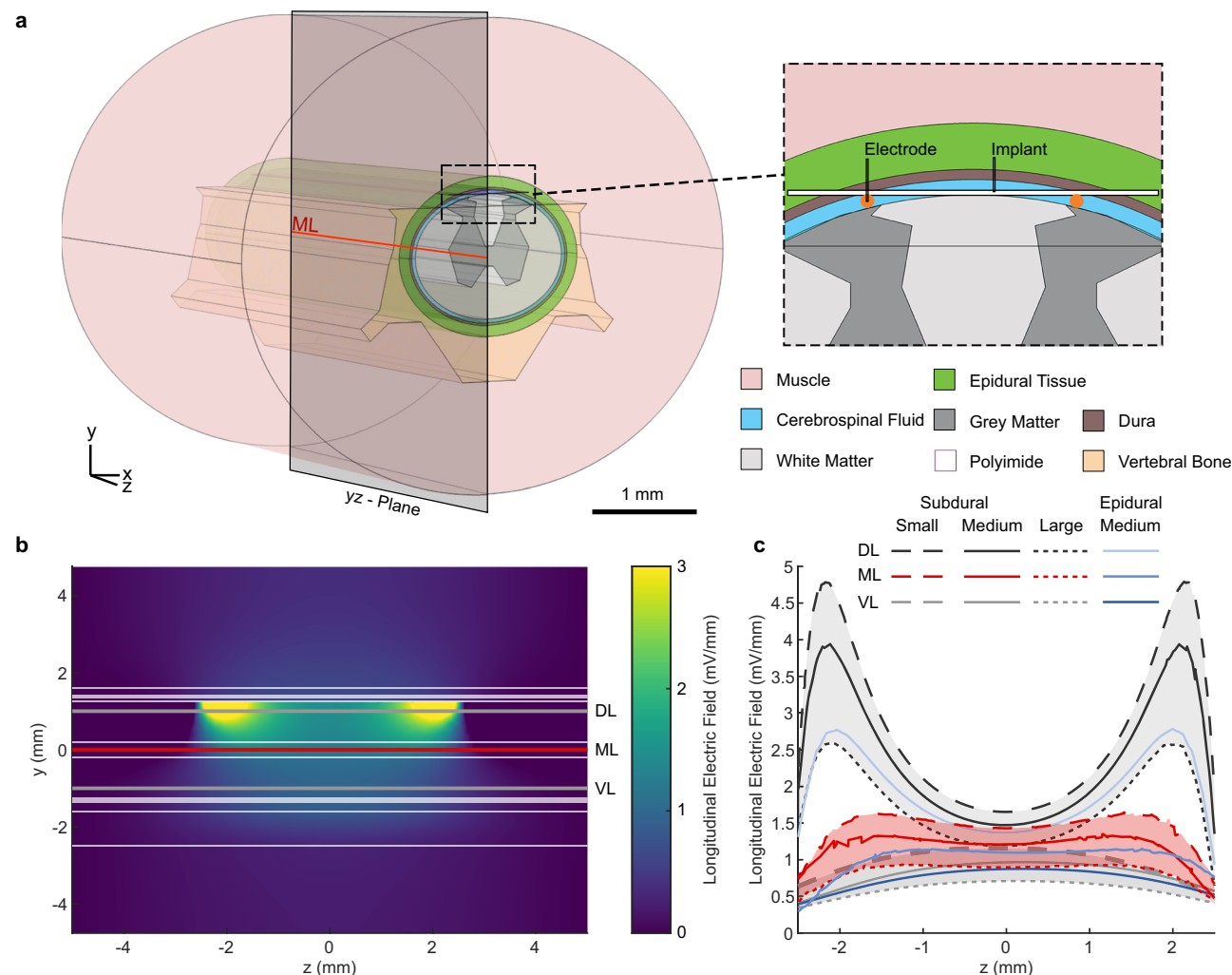

**Fig. 4 | FEM was used to estimate EF strength in the rat spinal cord and assess effects of cord size variations. a** The generated EF was estimated by applying the finite element method (FEM) to a model of the rat's spinal cord. **b** The longitudinal EF of the medium spinal cord is focused between the stimulation electrodes. **c** For the subdural implant, three different cord sizes were simulated (see dimensions in Table 2). At the midline of the simulated spine (ML) the longitudinal EF plateaus in the range of 0.8 to 1.6 mV/mm depending on the cord size over a distance of 4 mm, while it is weaker at the ventral line (VL, 1 mm ventral from ML) and stronger with peaks at the electrode positions at the dorsal line (DL, 1 mm dorsal from ML). The generated field strength decreases when moving the implant epidural.

window[32] (Fig. 5c). Post-stimulation CV showed higher peaks than pre-stimulation, likely due to stimulation-induced hydration of the SIROF (Fig. 5d, e). The 90 h continuous stimulation test applied greater electrochemical stress than in vivo conditions, where stimulation totals approximately 60 h over 12 weeks with recovery periods in between. Despite this increased stress, SIROF electrodes maintained their stability.

For long-term in vivo assessment of SIROF electrodes, we applied daily 1-h stimulation to a pair of electrodes on implant sample A under accelerated aging conditions (55 °C for 18 days, equivalent to 62 days at 37 °C[33]). Following this, a second electrode pair on implant sample A was stimulated under non-accelerated conditions (37 °C for 60 days). In each setup, one electrode served as the working electrode (WE) for 1 h, then the other, with four electrodes shorted as counter electrodes (CEs). Throughout the tests, the WE surfaces (Fig. 5f), EIS, and CV peaks remained stable (Fig. 5g, h; Supplementary Fig. 4b–d), indicating consistent electrode performance.

However, the CEs exhibited surface changes over time, with the typical SIROF fiber-like morphology becoming obscured by a denser material layer (Fig. 5f4, Supplementary Fig. 5). Importantly, CV and EIS measurements confirmed that the CEs remained functional, though an increased resistive component suggested a new surface layer was

deposited (Supplementary Fig. 4e, f). To further investigate this effect, we tested an additional set of electrodes on implant sample B, mimicking the in vivo stimulation by shorting two electrodes as CEs and stimulating one electrode as WE for 1 h daily over 60 days at 37 °C. Similar to the previous experiments, the electrodes remained functional, but this time the deposited layer occurred on the WE. Chemical analysis revealed that carbon was deposited (Supplementary Fig. 4g, details in Supplementary Note 1). This phenomenon was not further investigated in this work, as electrodes with deposited carbon remained functional, and nothing similar was seen on explanted electrodes.

## Electrode impedance increased in vivo, independent of their active use in treatment

To assess electrode functionality in the treated group, impedance spectra were recorded from the two active electrodes rostral to the injury. The four shorted electrodes at the caudal end of the implant served as the CEs for impedance measurements. Measurements were taken before implantation in PBS (in vitro), immediately after surgical implantation, and weekly thereafter. The measured impedance increased across the frequency spectrum over the 12 weeks of current-controlled stimulation (Fig. 6a), while the phase shift suggested

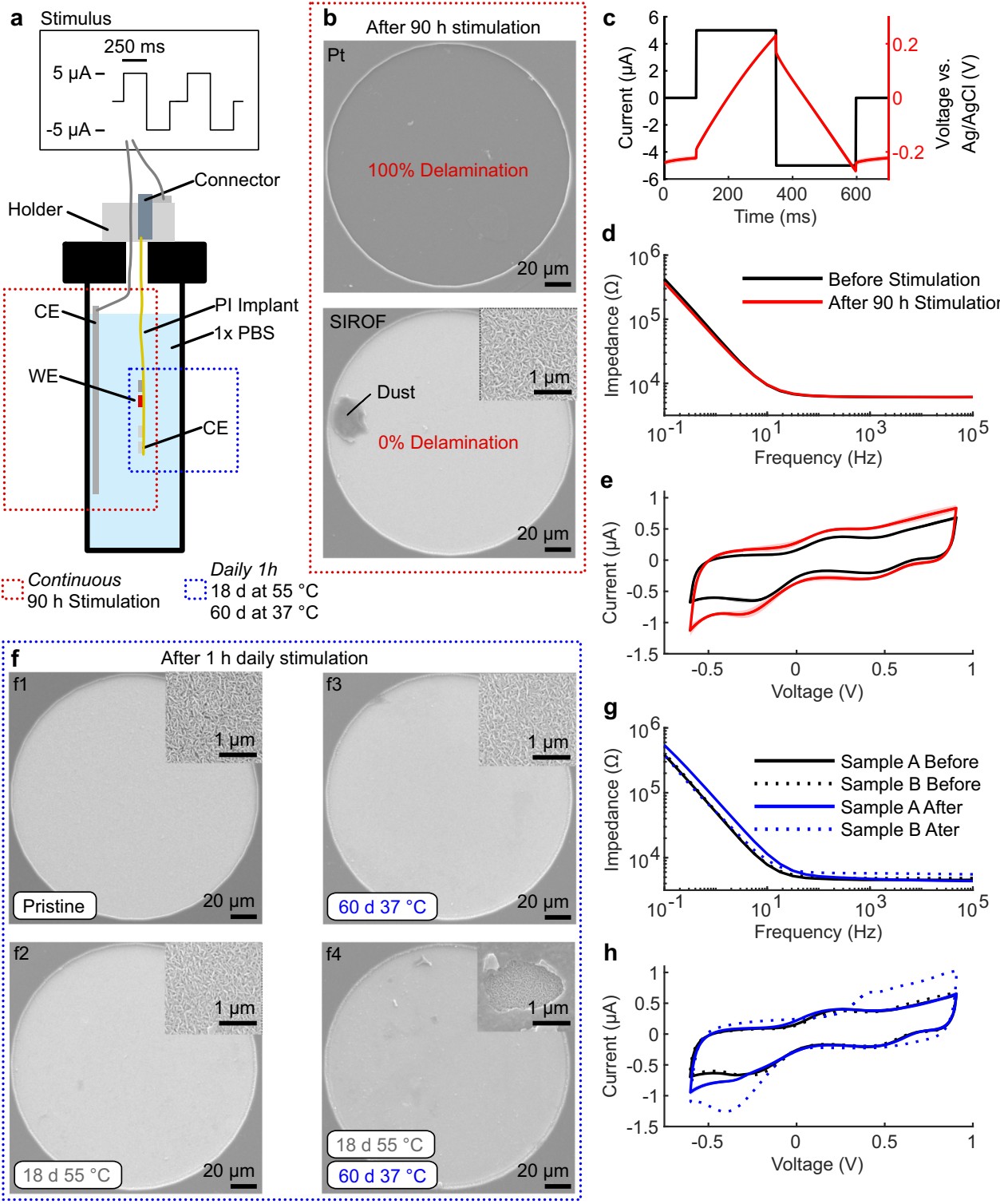

**Fig. 5 | The SIROF electrodes remained stable during in vitro benchtop testing.**
**a** The stability of Pt and SIROF-coated Pt was assessed during 90 h of continuous stimulation surpassing the total stimulation time in vivo (~60 h). In addition, the stability of the SIROF electrodes was tested by mimicking the in vivo treatment of daily 1-h stimulation for 18 days at 55 °C and for 60 days at 37 °C using the same implant body but different electrodes. **b** SIROF coating enhanced the stability of the electrodes. While the Pt electrode delaminated and dissolved during 90 h of continuous stimulation (Supplementary Fig. 4a), the SIROF electrode stayed intact. **c** The voltages reached during stimulation are within the water window. **d**, **e** Peaks in the CV and EIS indicate a functional SIROF electrode after 90 h of continuous stimulation (*n* = 3). The changes in the peaks of the CV after stimulation are likely

due to the hydration of the SIROF. **f** SIROF electrodes do not delaminate during the 1-h daily stimulation in accelerated aging regime and in vivo mimicking conditions. **f1–3** Representative images of SIROF electrodes, in pristine condition, and after daily 1-h stimulation for 18 days at 55 °C and for 60 days at 37 °C show that the electrodes remained stable. **f4** We observed a carbon layer deposited onto electrodes (Supplementary Fig. 4g, 5). **g**, **h** CV and EIS recordings of WEs which were stimulated for 60 days at 37 °C. The WEs stimulated consecutively (*n* = 2) against four shorted CEs remained stable (sample A), while the WE stimulated against two shorted CEs (sample B) remained functional although with carbon deposited at the WE.

increased capacitance at higher frequencies (Fig. 6b). Despite these changes, the impedance never reached open circuit levels, which would indicate electrode disconnection; instead, the EF treatment would have been achieved at a higher voltage, provided it did not exceed the stimulator's compliance voltage (16 V). Impedance readings from the two reserve (non-stimulated) electrodes on each implant (see Fig. 1a) showed similar trends over time (Fig. 6c). The difference between the final in vivo impedance magnitude measurement at 1 kHz for stimulated and reserve electrodes (i.e., electrodes not used for stimulation) was not significantly different from 0 (unpaired *t*-test, $p = 0.16$), indicating that stimulation did not contribute to the differences in impedance observed in vivo. Indeed, two devices that were removed intact at 3 and 4 weeks showed impedance returning to pre-implantation levels, after mechanical removal of encapsulating tissue, when tested in PBS (Fig. 6d, e; Supplementary Fig. 10a, b). We were unable to explant any other devices intact at later stages due to progressive tissue regrowth and encapsulation of the implant body.

### Electrode surfaces delaminated upon removal of the device

Although intact implant removal was not achievable at 12 weeks, the polyimide section interfacing with the spinal cord was successfully retrieved. During removal, we observed that some electrode surfaces (recording grounds and stimulation electrodes) delaminated from the polyimide and adhered instead to the spinal cord surface (Supplementary Fig. 6). For further investigation and quantification of this effect, we adapted the electrode status categorization scheme proposed in a previous study[34]. Using high-resolution images, each stimulation electrode was classified into one of five categories, ranging from Category 1 (pristine condition) to Category 5 (heavy delamination) (Fig. 7a, Supplementary Fig. 7). The key question was if the severity of delamination depended on the applied stimulus. The analysis covered the three groups of stimulation electrodes (all SIROF coated and 200 µm diameter): the electrodes that were stimulated as WE featuring the highest current density, shorted CEs which due to their larger combined area saw half of the current density, and electrodes that were never stimulated, referred to as full, half, and non-stimulated electrodes, respectively. Full, half, and non-stimulated electrodes exhibited similar delamination distributions, with 35%, 33%, and 25% of electrodes assigned to Categories 4 (more than half the area delaminated) and 5 (Fig. 7b), respectively. Interestingly, the full-stimulated electrode group had the highest number of electrodes in category 1, indicating that the severity of delamination is independent from the applied current density.

## Discussion

It has been known for more than 40 years that the application of pulsed EF promotes the bidirectional regeneration of axons after SCI. To date, nearly all in vivo studies have used epidural stimulation electrodes[16]. However, it has been reported previously that epidural electrodes require an order of magnitude higher current and produce significantly reduced EF strength and stimulation precision within the cord compared with subdural electrodes due to the requirement to penetrate through the meninges and CSF[19,20]. While previous studies have focused on humans, we analyzed the differences in EF strength between subdural and epidural placement in a rat spinal model. We found that this difference in rats is smaller due to their narrower subdural space[35], which minimizes the shift needed for an implant to become epidural and reduces its impact on EF strength. In humans, the larger subdural space provides more room for implantation but also increases the risk of complications, such as CSF leakage. It is thus too early to state what the ultimate solution will be for translation to patients. It is nevertheless self-evident that an implant in immediate contact with the spinal cord is more likely to reach efficient EF strength at lower currents, than one positioned epidurally, which only connects indirectly via the dura and the CSF-filled subarachnoid space. Although

implanting electrodes in the subarachnoid space requires more invasive surgery, advances in surgical techniques make the benefits of subdural devices a promising prospect for future medicine[18]. Therefore, the main objective of this study was to determine whether EF treatment could be safely delivered via subdural electrodes and whether it promotes regeneration and improves functional outcomes in a rat thoracic SCI model.

The EF treatment significantly improved the rat's locomotion. Treated rats showed enhanced coordination, paw position, and toe clearance in an open field from 4 weeks onwards compared to non-treated ones (Fig. 1e, Supplementary Video 1). The initially attenuated recovery observed in the first weeks may be attributable to a more severe injury caused by mechanical forces exerted during the physical connection of the stimulator, which was applied exclusively to the treated rats. Similar findings of connection-induced tissue damage have been reported in previous studies[36]. The treatment did not improve the outcome of the horizontal ladder task (Fig. 1g, h). Meanwhile, the impaction profiles suggested the injuries in these groups were the same (Fig. 1c). Furthermore, treated rats showed quicker withdrawal from a von Frey filament (Fig. 1f) than SCI controls indicating the recovery of touch sensitivity, which matches previous studies using low-frequency treatment in rats, dogs, and humans[3,6,14,37,38]. In a recent study, continuous epidural EF treatment (24 h/day) in conjunction with a T9 compression injury resulted in significantly higher BBB scores in treated rats from week 4 onwards compared to non-treated controls[37]. Additionally, increased protein levels of neurofilament, GAP43, and myelin basic protein were observed at 8 weeks post-injury. Previous work by the same group reported increased white matter preservation and neurofilament density, along with decreased astrocyte density at 4 weeks post injury, despite no improvement in BBB scores[29]. In contrast with these studies, we did not find evidence of increases in markers indicating neuronal/axonal structural integrity (B-tub) or recent axonal regeneration (GAP43) after 12 weeks. One possibility is that our 12-week endpoint may have exceeded the optimal window for detecting early regenerative changes, as the peak of cellular plasticity and regenerative signaling may have occurred earlier and subsequently diminished. Additionally, the contusion model used in our study generates a large cystic cavity, which presents greater challenges for detecting localized axonal changes compared to the compression injury model, which creates a more focal lesion with minimal tissue loss. These factors may have contributed to a more inhibitory microenvironment at the chronic injury stage, potentially limiting the capacity for axonal regrowth and obscuring measurable treatment effects at this later time point. Another possibility is that EF may facilitate functional recovery through neuroprotective effects or alternative mechanisms that do not directly involve axonal regeneration. For example, another study using continuous epidural EF treatment in a rat T9/T10 contusion model showed improved survival and differentiation of endogenous neural precursor cells after 5 weeks[39], and other studies demonstrate that EF treatment in animals can reduce reactive astrocytes and demyelination[40].

In this work, the estimated EF strength at the lesion was higher than in previous rat studies[39,41]. Previous histological data indicated that the untreated lesion extended up to 3 mm along the caudal-rostral axis and reached the ventral outline of the cord after 12 weeks[26]. We positioned stimulation electrodes 1 mm outside the lesion in healthy tissue, as this separation improved EF uniformity within the lesion[42]. The model did not include the contusion injury, as the effect on tissue conductivity is uncertain[43,44]; reduced conductivity would increase EF strength, while increased conductivity would have the opposite effect. Using healthy tissue conductivities is thus a reasonable compromise. Another limitation is the exclusion of the foreign body response, which may lead to inflammation and fibrotic tissue formation around the implant[45], however, previous research indicates that this minimally affects therapeutic EF in white and gray matter[42]. FEM-based analysis

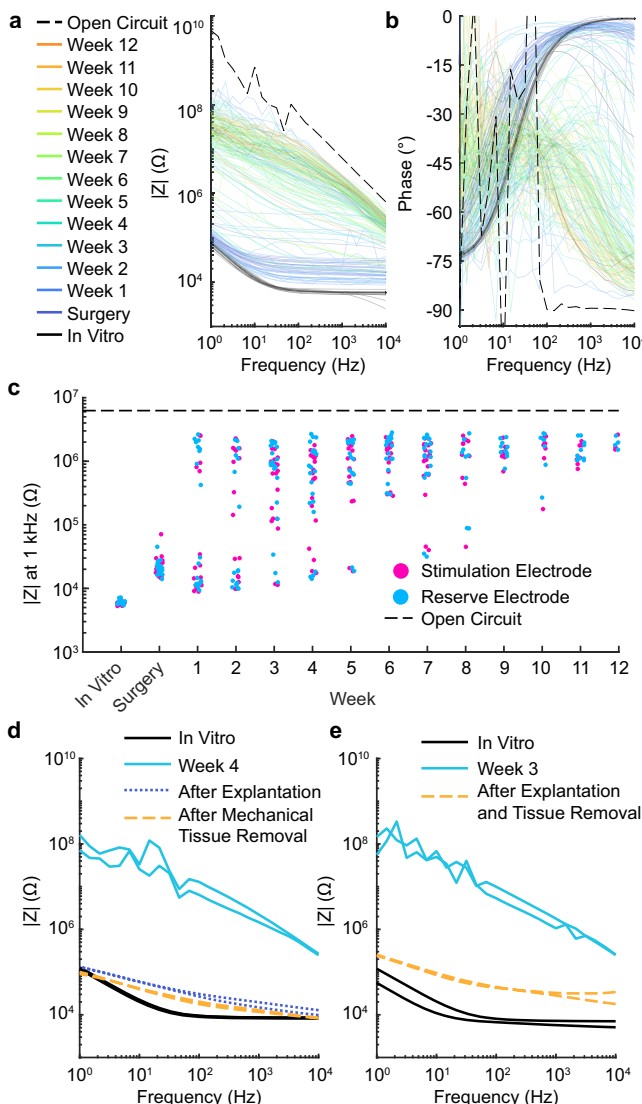

**Fig. 6 | Impedance of the electrodes in treated rats increased steadily in vivo, but this rise was unrelated to the delivery of treatment and reduced upon tissue removal in recovered implants. a** Impedance magnitude and **b** phase for active electrodes used for stimulation are shown over 12 weeks, indicated by the color scale, with the dashed red line representing the open circuit level. See Supplementary Fig. 9 for reserve electrodes. **c** Impedance at 1 kHz illustrates a steady increase throughout the 12 weeks for both stimulation and reserve electrodes, remaining below the open circuit level expected for a broken channel or electrode. **d, e** In vivo and in vitro measurements from two explants that were removed intact at 4 and 3 weeks, respectively. Both explants showed impedance approaching pre-implementation levels after mechanical removal of encapsulating tissue.

showed that, regardless of natural variability in spinal cord size, assuming the size of the untreated lesion, the injured tissue was exposed to a minimum longitudinal EF of 0.6 mV/mm, with regions dorsal to the midline receiving over 1 mV/mm (Fig. 4c). The relationship between EF strength and SCI regeneration remains poorly understood, complicating predictions about the impact of variations in cord size, injury, or foreign body response on treatment outcomes. Nonetheless, the simulated EF strengths (0.7–1.6 mV/mm) at the cord center exceed those shown to induce positive biological effects in earlier studies (0.14–0.6 mV/mm)[6,14,38,39,41], suggesting that any unmodeled factors would need a significant influence to alter the treatment results. Prior research applying a 0.5 mHz stimulus over longer daily durations estimated EF strengths around 0.4 mV/mm[39,41], while other

studies did not estimate EF strength[29,37]. Given the higher EF strength and the first implementation of a subdural thin-film implant for delivering EF treatment, we were especially interested in evaluating if there was a stimulation-induced foreign body response. There were no signs of increased microglia or astrocyte activity in rats treated with EF compared with the non-treated group, suggesting the daily treatment was well tolerated and it may be possible to deploy even higher EF strengths in the future or for much longer periods (Fig. 2). This is an important aspect of the work, as biocompatible stimulation without generation of toxic by-products relies on the fact that current and duration stay below certain thresholds, and we were able to demonstrate that the combination of reduced current and revised electrode materials made this possible.

The stimulation electrodes were stable during in vitro tests in PBS when the appropriate electrode materials were used. SIROF electrodes showed no signs of dysfunction or delamination even after continuous stimulation exceeding the total in vivo treatment time, under accelerated aging, and conditions mimicking the in vivo protocol (Fig. 5, Supplementary Fig. 4). One deviation between in vitro and in vivo tests was the deposition of a predominantly carbon conducting layer on some electrodes. Since this was absent in vivo and did not affect electrode function, we did not investigate further (Details in Supplementary Note 1). In comparable studies the implant's functionality was analyzed at the end-point[14], additional recording electrodes were implanted[29], or no analysis was reported[39,41]. We here complemented such analysis with an in vivo assessment of the implant in situ using EIS. Our aim was to utilize the recorded impedance spectrum to progressively evaluate foreign body response and the status of the active electrodes[46]. In vivo impedance of both active and reserve electrodes increased over time but importantly, remained within functional limits. The rise in impedance could be attributed to various factors, including tissue encapsulation, connection line failure, electrode delamination, or insulation issues, potentially in combination. Other studies have shown that similarly thick insulating polyimide sustains similar and longer periods in vivo than tested in this work[47–49]. High-resolution images of implants from the treated group show that connection lines and the polyimide insulation were intact (Supplementary Fig. 13). Due to the inability to remove implants intact at 12 weeks, connection line status could not be evaluated electrically in treated animals. However, implants retrieved at weeks 3 and 4 showed that, following mechanical tissue removal, electrode impedance returned to lower levels, suggesting that increases observed in vivo were partly due to implant encapsulation and biofouling (Fig. 6d, e; Supplementary Fig. 10). For implants after 12 weeks, high-resolution image analysis revealed that approximately one in ten stimulation electrodes showed substantial delamination (category 5), indicating non-functionality (Fig. 7b). Since delamination was absent during in vitro testing, it likely resulted from the additional mechanical stress during animal movements or explantation, as electrode material was found adhering to the spinal cord (Supplementary Fig. 6). It should be noted, that residual material from the electrodes in the sheath surrounding the implant were reported for commercial devices and are an acceptable risk[50]. In a recent study, the delamination pattern of comparable electrodes was investigated, and a similar finding of explantation or mechanical stress-induced delamination was made[34]. These results suggest that the electrodes effectively delivered stimulation over the treatment period, with some individual variation where it is not possible to fully exclude earlier interruption due to delamination. Future studies should incorporate additional methods (e.g., monitoring the current or testing the threshold to evoke motor function) to assess implant functionality in situ or utilize coatings that reduce encapsulation, facilitating explantation. Delamination may be addressed by implementing additional adhesion layers between polyimide and

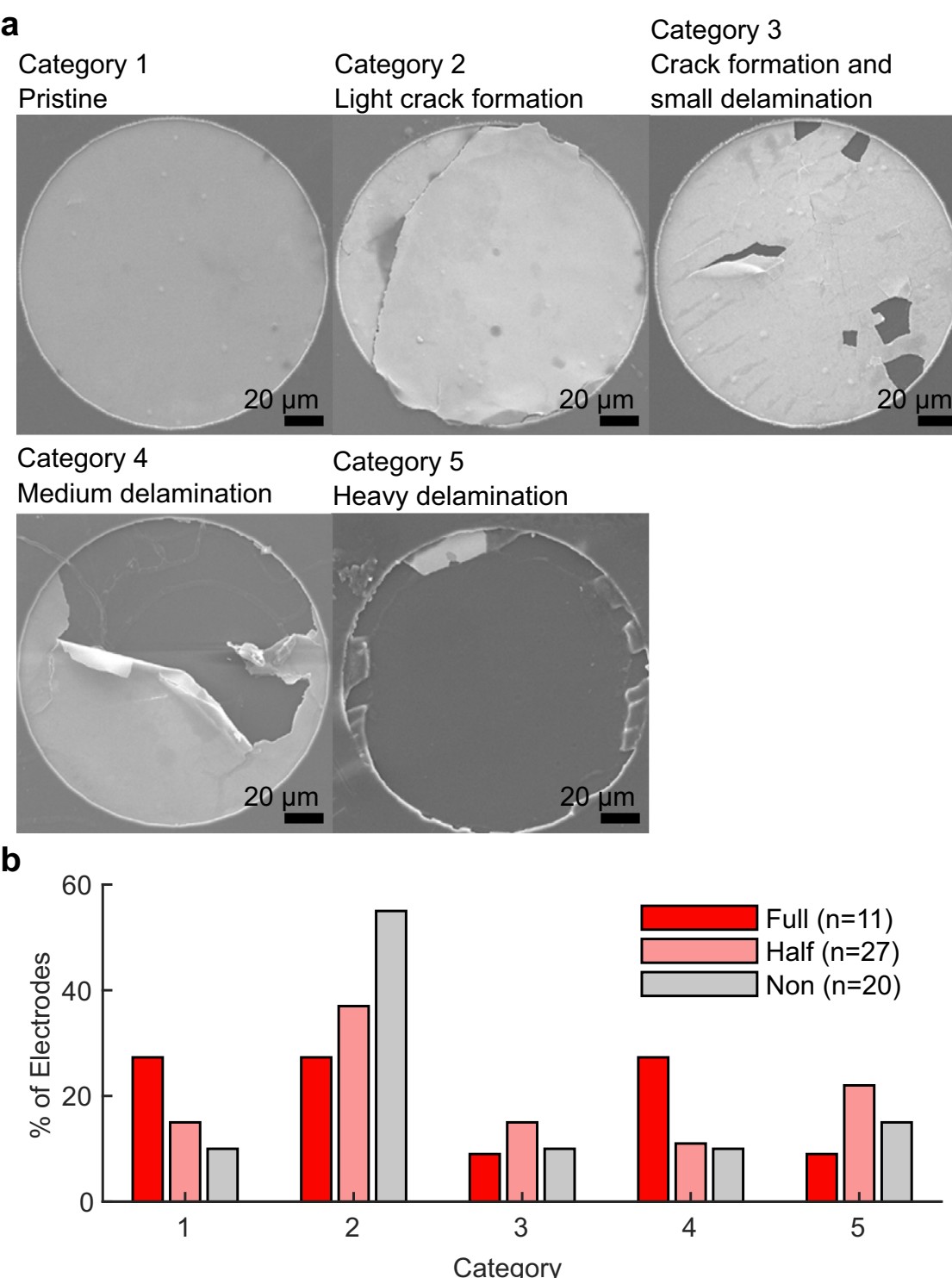

**Fig. 7 | Delamination of electrodes occurred during explantation and was categorized into different severities. a** Categories were **a1** pristine, **a2** light crack formation, **a3** crack formation and small delamination, **a4** medium delamination and **a5** heavy delamination. **b** The distribution of electrodes across each status category, based on the current density received.

platinum (e.g., silicon carbide)[51]. Importantly, a therapy based on the concept presented here would not necessarily require a permanent implant. While it is too early to determine, the ideal scenario would be if stimulation over several months were sufficient to produce lasting beneficial effects. Further research is needed to investigate this possibility.

This study has several methodological limitations, which we aim to address in future research. We investigated the feasibility of subdural stimulation but did not include a group with epidural stimulation, which prevents a direct comparison of the efficacy of both methods. While we did not find clear evidence of regeneration, the transient expression patterns of GAP-43 and other regeneration-associated markers, such as brain-derived neurotrophic factor, may have declined by the 12-week time point. To overcome this limitation, future work will incorporate brain and spinal injections of viral tracers to label specific supraspinal motor and sensory tracts, respectively.

Unlike regeneration markers, these tracers provide timeline-independent labeling, allowing for a more direct assessment of axonal extension and connectivity. To address the challenges of thin-section histology with the large lesion cavity in the contusion model, we will combine viral tracers with volumetric imaging of thicker sections or whole spinal cords using confocal and light-sheet microscopy. The von Frey task indicated a quicker withdrawal by treated rats after just one week. While this could reflect an early effect of the EF treatment on the acute stage of injury, it could also be influenced by reduced sample size at this time point due to non-plantar paw positioning. In the future, we will begin this task at week 2. In addition, at several timepoints, treated rats paw withdrawal was quicker than non-injured rats. This may suggest a degree of hypersensitivity associated with recovery, potentially affecting their paw placement accuracy on the error ladder[52]. However, it remains unclear due to high variability of this task in uninjured rats[27]. In future, we plan to perform additional thermal and cold sensory tasks as well as place-preference tasks shown to be sensitive to neuropathic pain. We show that subdural insertion of the implant using gelfoam to form a hemostatic seal is tolerated well in rats; we did not observe any motor impairments in the no-injury group. However, a different approach would be necessary in larger mammals to prevent CSF leakage.

This study demonstrates that subdural stimulation enhances the recovery of rats after SCI. However, the role of pulse modalities remains unclear. Despite significant differences in treatment protocols, including daily duration, EF strength, and overall length, the resulting locomotion recovery in this study was comparable to previous findings. Thus, further exploration of a wider EF strength space is essential to determine optimal treatment conditions, something that will be enabled by progress in implant technology and where this paper is an important first step. Increasing the magnitude of the generated EF strength might enhance the treatment's success. The applied EF strength in cell culture to promote axon growth is usually a magnitude larger than what is considered in vivo. Previously, a minimum required field strength of 5 mV/mm to promote axon regrowth was suggested[53]. Other studies have concluded that the time-averaged field intensity rather than field strength alone, is the axon growth-promoting factor[54–56]. In both cases, the stability of the electrode material is the decisive factor, as increasing the current to enhance the EF, or extending the pulse width, ultimately increases the electrochemical stress on the electrodes. To this end, we recently developed a version of the implant with larger poly(3,4-ethylene dioxythiophene) based electrodes, which will be used to investigate the effect of higher charge delivery per pulse in the future.

## Methods

### Implant fabrication
The implants were fabricated in the cleanroom facility at the University of Freiburg (RSC, ISO 4, according to ISO 14644-1) following similar fabrication steps as in previous works[49,57]. First, polyimide (U-Varnish-S, UBE Industries Ltd., Japan) was spun onto 4-inch silicon wafers at 4500 rpm to achieve a thickness of 4 μm after annealing. Then, the image reversal photoresist AZ 5214 E [EU] (MicroChemical GmbH, Germany) was used to pattern connection lines and pads. After O2 plasma activation for 30 s (100 W, STS Multiplex ICP, SPTS Technologies, United Kingdom), 100 nm Pt was evaporated (Univex 500 Electron-Beam Evaporator, Leybold GmbH, Germany) followed by a liftoff in acetone. Subsequently, the image reversal photoresist was used to pattern the electrodes. Then, first 100 nm Ir followed by 700 nm IrOx was sputtered (100 W DC, UNIVEX 500, Leybold GmbH, Germany). After liftoff in acetone, the wafers were O2 activated for 30 s and a second layer of polyimide was spin coated at 4500 rpm and cured. The positive resist AZ 10XT (MicroChemical GmbH, Germany) was used as etch mask during reactive ion etching in O2 plasma

(100 W). After stripping the resist in acetone, implants were peeled from the silicon wafers using tweezers. The implants were soldered (LFM-65W TM-HP, Almit Ltd., England) onto custom PCBs. The electrodes on the implants were then shorted and 500 activation CV cycles (−0.6 to 0.9 V vs. Ag/AgCl, 100 mV/s) in 1× PBS (P3813, Sigma–Aldrich) were performed in a three-electrode setup with an Ag/AgCl reference electrode (Ag/AgCl, BASI, USA), and a stainless-steel counter electrode (-20 cm²) using an autolab potentiostat (PGSTAT 302 N, Metrohm Autolab B.V., Germany). Each electrode was individually characterized with EIS and CV in the same electrochemical setup to assess functionality.

### Animals and housing
Thirty-three 2–3-month-old female Sprague Dawley rats were obtained from the University of Auckland animal facility (VJU SBS). All work complied with the University of Auckland Animal Ethics Committee Guidelines (AEC22644) and the New Zealand Animal Welfare Act 1999. Weights ranged from 210 to 290 g at the time of surgery and increased to 236–368 g at the end of the experiment, 12 weeks post operation. Rats were housed in a dedicated colony/behavioral room on a 12:12 light/dark cycle with rat chow and water available ad libitum throughout the experiment. They inhabited large cages (44 by 34, 24 cm high) with sawdust bedding and were given wooden chew sticks and shredded cardboard slivers for nesting. They were group-housed (4–5 per cage) for two weeks prior to surgery during which they were extensively handled and habituated to behavioral tasks (see below). Post-surgery, rats were individually housed in the same type of cages for 12 weeks. Once bladder function was spontaneous, cages were changed once/week.

### Surgical procedures
Animals were anesthetized with isoflurane, temperature monitored and placed on a heating pad and given subcutaneous antibiotics (Baytril 25 mg/mL, 0.04 mL per 100 g) and analgesics (Metacam 5 mg/mL, 0.03 mL per 100 g; buprenorphine 0.3 mg/mL, 0.1 mL per 100 g; bupivicaine injected along incision site 2.5 mg/mL, 0.1 mL per 100 g). A 6–7 cm incision was made and a laminectomy was performed on spinal processes T10, T11, and T12, exposing the spinal cord segments T13, L1, L2 and L3. Small incisions in the dura were made near either end of the exposed spinal cord in either hemisphere using a 27.5 G needle. Guidance catheters (Alzet, 0007741) attached to each arm of the implant via threaded sutures (Polyamide, 7.0) were inserted below dura in either hemisphere at the T13 end and carefully propelled along the surface of the spinal cord[58], exiting at the L3 end. The sutures were used to insert the implant arms and position them so that the stimulation electrodes were in contact with the spinal cord. An Infinite Horizons impactor was then used to deliver a 175 kilodynes impact injury using a 2.5 mm diameter tip to the border of spinal segment L1/L2 and carefully centered within the laminectomy between the two sets of stimulation electrodes (Supplementary Fig. 1A). The impact tip is then raised -5 mm above the spinal cord and triggered via software (IH Spinal Cord Impactor v5.0) to deliver the impact directly above the subdural thin-film implant; performing the impact injury with the implant in place did not alter the impact mechanics or affect electrode function (Supplementary Fig. 12). A sensor on the impactor measures the force (kDynes) and duration (ms) of each impact (Supplementary Fig. 1b, c). This injury severity provides an intermediate level of recovery allowing sufficient room for treatment effects. A more severe injury was not induced because weight bearing and proper plantar foot placement were essential for the error ladder and von Frey tasks. Additionally, severe injuries are associated with greater health and welfare risks over extended (12 week) recovery periods, and translating a viable treatment is more likely to first target moderate injuries before advancing to more severe injuries. After impaction, the tips of each implant arm were sutured to the nearby T13 spinal process (Polyamide,

7.0). A thin piece of absorbable gelatin sponge (Pfizer Gelfoam) was used to form a hemostatic seal in the space above the exposed spinal cord and the muscle layer above sutured closed using an absorbable suture (ChomicGut, 4.0). At the rostral end, the implant was connected to a small PCB with a 36-channel Omnetics female nano-strip plug housed in a 3D printed backpack (see ref. 17). The base of this backpack consisted of five splayed feet attached via sutures to an oval piece of surgical mesh. Feet and mesh were placed in a subcutaneous pocket and sutured to the deeper muscle through the surgical mesh using non-absorbable nylon suture (Polyamide, 4.0). Over several weeks, the mesh provided a scaffold for tissue regrowth resulting in a strong anchor ensuring minimal risk of the implant or backpack becoming detached. The skin was sutured closed tightly around the base of the backpack (Silk, 4.0), and the hind-limb claws were carefully trimmed under the microscope to reduce damage caused by scratching. Post-operatively, animals were given analgesics (Metacam 5 mg/mL, 0.03 mL per 100 g; buprenorphine 0.3 mg/mL, 0.1 mL per 100 g) and supplementary fluid (saline, 3 mL) for 3 days. Bladders were voided manually by experimenters if required in the early morning, midday and evening each day. Subcutaneous antibiotics (Baytril 25 mg/mL, 0.04 mL per 100 g) were given daily and cages/bedding were replaced every two days until rats were able to urinate spontaneously.

## Experimental groups

Thirty-three rats were implanted, 23 of which also received a contusion injury with the remaining 10 designated as non-injured group. Five subjects were identified as statistical outliers and were removed from the study on the basis of post-surgical BBB scores that varied more than two standard deviations from their group mean (treated group, $n = 2$; non-treated group, $n = 3$); see Supplementary Fig. 1e, f. Two of the outlier rats were perfused in weeks 3 and 4 to examine electrode impedance of the explanted device. One rat in the non-treated group was euthanized after week 8 due to a foot injury followed by infection, and was excluded from behavioral assessments from that point onward. After 12 weeks, the spinal cords from six treated and three non-treated rats were processed for histology. Final group sizes were: No injury, $n = 10$; treated $n = 8$; non-treated $n = 10$.

## Electric field treatment sessions

Starting the day after surgery, animals received 1-h pulsed electric field treatment for 7–11 subsequent days, then on weekdays only (5 days/week) for 12 weeks. Each rat had 62 treatment sessions over 84 days. Animals were placed on a comfortable towel-lined platform that sat within a walled bucket (Supplementary Fig. 2a). They were gently restrained by an experimenter, while another carefully plugged in (or unplugged) the implant using an overhanging 1.2 m Omnetics cable plugged into a commutator (SenRing, M220A-36). Four such cables and commutators could be simultaneously plugged into four HS32 Headstages connected with an ME2100 electrophysiology system (Multi Channel Systems) and desktop computer (Supplementary Fig. 2b). Stimulation was initiated using Multi Channel Systems Experimenter Software; in each headstage, separate green and blue stimulators provided current-controlled pulsed stimulation to an electrode on either arm of the implant. Four stimulation electrodes on the rostral side of the injury were shorted together in the cable to enable biphasic pulses (Supplementary Fig. 2c). Shorting four stimulation electrodes and using them as counter electrodes was necessary because the stimulator permits only a single ground pathway. The pulse parameters were +5 µA for 250 ms, 0 µA for 1 ms, −5 µA for 250 ms, 0 µA for 100 ms, then repeated for 60 mins (Supplementary Fig. 2d), delivering ~5990 biphasic pulses in each session. We carefully monitored each animal during initial stimulation sessions to ensure there were no unintended motor responses.

The safe charge injection capacity for SIROF with alternating current with pulses of durations in the range of 100–500 µs was reported between 4 and 5.8 mC/cm[2,59]. The safe charge injection limit represents how much charge can be delivered before the onset of water electrolysis, which results in cytotoxic stimulation by-products and pH changes. If we follow the 15 min pulse width used in prior studies and assume previously reported safety windows apply, the 200 µm diameter SIROF electrodes then only allow for a current in the range of nA's, which would generate an insignificant EF. Instead, we aimed to deliver a 250 ms biphasic pulsed EF, which previously showed promising results in vitro[24]. Using µs pulse-widths mainly takes advantage of capacitive effects which do not elute high concentrations of stimulation by-products into the tissue[59]. Since the pulse width applied in this work is more than 1000 times longer than what was tested for the safe charge injection limit, faradaic reactions contribute to the charge transfer, which have the potential to generate toxic concentrations of stimulation by-products and to degrade the electrode if not thoughtfully selected. In a previous study, we found that SIROF electrodes stimulated with 10 µA/cm[2] reached a plateau in voltage during direct current stimulation after 19 min, indicating that irreversible faradaic reactions dominated the charge transfer beyond this point, with oxygen generation occurring at the anode after 20 min[23]. The 10 µA/cm[2] for 19 min equals a charge density of 11 mC/cm[2]. In this work, we decided to settle for the current density of 4 mC/cm[2] yielding a current amplitude of 5 µA. Therefore, the applied stimulus in the current study is inside the previously found safe charge injection limit for short µs-pulses and well below the limit for the onset of water electrolysis at the anode for SIROF electrodes during direct current stimulation.

## Open field and BBB-scoring

The BBB locomotor rating scale was used to measure right and left hind-limb motor function. Animals were placed in a circular open field (100 cm diameter white wooden floor with 20 cm Perspex walls) in the colony room. Prior to surgery, rats were placed in the open field with cage-mates for 10 min on two consecutive days; on the third day they went in individually and a baseline session was recorded. Post-surgery, rats were tested on day 1, 3, 7, and then weekly for 12 weeks. Each session, rats were placed in the center of the open field and then allowed to freely move around for 5–8 min depending on the amount of movement exhibited. If the animal stayed in one place for ~30 s they were carefully picked up and placed in the center. BBB scores were determined from videos of the sessions by a single experimenter (BH) blinded to the condition of the rats during scoring; DeepLabCuts was used to blur out labeling on the backpacks that identified the rats as individuals, and video names were replaced with random words. The functionality of each hind limb was scored from 0 (total paralysis) to 21 (normal movement) and the scores for the right and left hind limbs averaged. Segments of each video were found in which the animal moved around the outside or across the middle of the arena at a consistent speed for approximately five or more step cycles. This allowed a more accurate assessment of front-hind limb coordination and ensured animals were moving at a similar relative speed[60,61]. Examples of open field locomotion are shown in Supplementary Video 1.

## Electronic von Frey task

Mechanical sensitivity of the hind paws was tested weekly using an electronic von Frey Plantar Aesthesiometer (Ugo Basile), which allows automated application of a filament using a ramping force (set at 2.5 g/s, with maximum 50 g after 20 s). Prior to surgery, rats were habituated to the compartments (5–6 rats at a time) for 20 min on two consecutive days. On the second day, rats were habituated to the presence of the movable touch stimulator, which was moved around below the mesh testing grid floor, and on the third day, a testing session was administered. At the start of each session, rats were left alone for 15 min or longer if required for them to adjust to the testing

compartment and become settled (15–25 min). The filament was then applied to each hind paw three times each with an inter-trial interval of 3–4 mins, alternating between the left and right paws. At weeks 1 and 2 post injury, hind paws were not always plantar with the mesh floor; additional measures were attempted at the end of the sequence of testing, but if not possible the measures were not attempted.

## Horizontal ladder task

Starting two or three weeks after surgery (when animals are weight bearing during stepping), animals performed at least 3 traversals of a 1-m horizontal ladder once a week. The irregularly spaced steel rungs of the ladder were suspended between a white wooden wall behind and perspex wall in front to create a 10-cm wide corridor. A 45° mirror positioned below the ladder allowed a Hero7 Silver camera (GoPro) to record the animal's foot placement from the side and below. At one end of the ladder there was a perspex start box, and at the other end a black wooden box contained a Perspex ramp leading into a plastic tray with sawdust bedding. Prior to surgery, rats were placed in the apparatus with a different rung patterned ladder and with cage-mates for 10 min on two consecutive days and allowed to freely explore; on the third day they went in individually and a baseline session was recorded. In this baseline session and all subsequent weekly sessions, the same rung configuration was used. We decided that the short weekly exposure would allow minimal learning of the configuration, and this was preferable to changing it, which would make comparison of the weekly sessions (and recovery) problematic. Each session, animals were placed in the start box and allowed to run across to the goal box. Rats were then allowed to traverse the ladder in the other direction, however, if they remained in the goal box for over 30 s, they were picked up and replaced in the start box until they had performed 4–5 traversals of the ladder. Sessions were scored from videos by a single experimenter (CC) blinded to the condition of the rats during scoring; DeepLabCuts was used to blur out labeling on the backpacks that identified the rats as individuals, and video names were replaced with random words. Each paw placement on the ladder (front-right, front-left, hind-right, hind-left) was scored with a 0 for normal placement, 1 for a misplacement, and 2 for a slip resulting in loss of weight bearing.

## Histology

After 12.5 weeks, rats were euthanised with sodium pentobarbitone (100 mg/kg) and intracardially perfused with 0.9% saline (300 mL) followed by 4% paraformaldehyde (PFA, 300 mL, in 0.1 M phosphate buffer, pH 7.4). A 1 cm section of spinal cord, centered around the injury was excised and post-fixed in 4% PFA for 4 h at 4 °C, followed by 20% sucrose for 24 h, and stored in 30% sucrose with 0.1% azide. Sections were embedded in OCT and stored at −80 °C before cryosectioning sagittally at 20 μm using a Leica CM3050S. Every other slice (-1000 μm) was mounted on positively charged slides (Supplementary Fig. 8a). Sections were blocked with 3% normal donkey serum (NDS), incubated overnight with primary antibodies in 1% NDS/PBS (Supplementary Fig. 8c), followed by secondary antibodies for 2 h (Supplementary Fig. 8d). Hoechst 33342 (1:10,000, ab228551) was used as a counterstain. Slides were cover-slipped with ProLong Gold and imaged using a Zeiss Z2 Axioimager with a MetaSystems VSlide scanner operated via MetaFer (V3.12.1) software. Fluorophores used included Alexa Fluor™ 647, 594, 488, and Hoechst 33342, with consistent exposure settings. Images from 3–5 tissue sections per cord were cut into 2500 μm × 2500 μm sections and analyzed in Fiji software. The image processing pipeline is shown in Supplementary Fig. 8b. Dorsal and ventral sections were defined from the lesion center and validated manually. Binary images were created for each color channel (B1), and background subtraction (500 pixels) isolated neuronal/axonal structures (B2). Control histograms set thresholds to remove background (B3), creating stain fluorescence masks (B4). Fluorescently labeled areas were normalized against Hoechst-positive cells to control for tissue size variations.

## Computational modelling

The EF distribution was simulated in COMSOL Multiphysics® software (version 5.3). Atlas segments were used to determine the shape of the gray matter and the vertebral bone[30]. The white matter was modeled as an elliptical cylinder. The width and height of the white matter was varied according to the natural spinal cord size variability of L1 in rats reported in previous work[30] resulting in three different simulated cord sizes namely small, medium, and large. For the differences in white matter width and height we used the standard deviations at L1 position. At L1 mainly the volume of white and not gray matter varies[30]. In these three cord sizes we also varied the width of the CSF in the range of 0.05 to 0.15 mm, because we hypothesized that due to its high conductivity the volume of CSF surrounding the stimulation electrodes also influences the generated field strength in the spinal cord. As we have not found reference data based on anatomical measurements for CSF thickness, we adapted and expanded the thickness range from a prior simulation study[62]. The conductivities and the varied dimensions of the three simulated models are shown in Tables 1 and 2. For all models, the dura, and epidural tissue were modeled as layers surrounding the white matter with widths of 0.05, and 0.15, respectively[62,63]. The surrounding muscle body had a width of 10.65 mm and height of 9.5 mm. Four rectangular electrodes (Pt, two rostral, two caudal, 177 × 177 μm) were positioned in the CSF with 5 mm separation in rostral – caudal and +- 0.55 mm from the midline of the cord. On top of the electrodes an 8 μm thick, 1.8 mm wide polyimide sheet was modeled. 5 μA was applied to each of the rostral electrodes while the caudal electrodes served as ground in a stationary electric current study. For the epidural implant, the model of the medium-sized cord was kept as is, just the implant was shifted by 0.125 mm in y-direction to lay epidural. The model was validated by comparing the relative field strength at the dorsal, central, and ventral positions to the relative measured field strengths in rodents stimulated with 10 mm separated epidural disk electrodes[43]. In rodents, at the point between the electrodes, the measured central and ventral field strengths are approx. 80% and 60% of the dorsal field strength, respectively. In the medium size model considering the subdural implant, the central and ventral field strengths are 82% and 65% of the dorsal field strength, respectively. In addition, when comparing simulated and measured field strength, the curves at the dorsal region were comparable with the field strength being maximum close to the position of the

## Table 1 | Tissue conductivities

| Tissue | Electrical conductivity [S/m] | Ref |
|---|---|---|
| CSF | 1.79 | 64 |
| Gray matter | 0.23 | 65 |
| White matter | 0.6 | 65 |
| Dura | 0.03 | 65 |
| Bone | 0.02 | 66 |
| Muscle and epidural tissue | 0.2 | 53 |
| Polyimide | $6 \times 10^{-15}$ | 67 |

## Table 2 | Cord size variations

| Tissue/cord size | Small | Medium | Large |
|---|---|---|---|
| Cord width [mm] | 3.07 | 3.25 | 3.43 |
| Cord height [mm] | 2.35 | 2.5 | 2.75 |
| CSF thickness [mm] | 0.05 | 0.075 | 0.15 |

electrodes and then notably decreasing at the point equidistant from the electrodes.

### Benchtop voltage transients and long-term stimulation testing

Voltage transients were recorded in three-electrode setup using the autolab potentiostat (Nova 2.1.6). Before recording the voltage transient, electrodes were stimulated (+5 μA for 250 ms, 0 μA for 1 ms, −5 μA for 250 ms, 0 μA for 100 ms) for 10 min for the voltage excursion to reach stable state. Three stimulation electrodes on three different implants were used for voltage transient recordings. Three types of long-term in vitro stimulation tests were performed in 1× PBS: 90 h continuous stimulation, 18 days 1 h daily stimulation in accelerated aging regime, and 60 days 1 h daily stimulation at in vivo conditions. For all tests, we used an implant design that was similar to the in vivo implants but had additional fenestration, which does not interfere with their electrochemical performance. In all tests, the pulse parameters were the same as in vivo; +5 μA for 250 ms, 0 μA for 1 ms, −5 μA for 250 ms, 0 μA for 100 ms, then repeated for 60 min. The pulse was applied with the autolab potentiostat. For the 90 h continuous stimulation, we first tested one implant without SIROF coating. We applied the stimulus to one 200 μm diameter Pt electrode. We then tested the 90 h continues stimulation on one SIROF-coated electrode on three independent implants. Each implant together with a stainless-steel mesh (~21 cm²) serving as counter electrode was mounted into a 50 ml tube (Centrifuge Tube, Conical Bottom, VWR) by drilling a small hole through the lid. The tube was then filled with 1× PBS and sealed with Blue Tack (Bostik, France). After termination of the experiment, the stimulated electrode on each implant was characterized in fresh PBS by EIS (4 points per decade, 100 mV peak to peak amplitude) and CV (−0.6 V to 0.9 V, 100 mV/s) in a three-electrode setup with an Ag/AgCl reference electrode and a stainless-steel counter electrode (~20 cm²). For the stimulation mimicking the in vivo treatment, one implant was employed. The implant was again mounted into a 50 ml tube filled with 1xPBS and secured via a 3D printed part (Gray Resin, Form 3B+, Formlabs Inc., United States) at the top of the lid. The hole in the lid was then sealed with silicone (Dow Silicone 734, Dow Chemical Company, United States). The tube was placed inside an oven (FD-S 56, Binder GmbH, Germany). In the first experiment we assessed the stability of the SIROF electrodes in accelerated aging regime at 55 °C for 18 days, which according to the "10-degree rule"[33] corresponds to 62 days at 37 °C. Each day, 1 h of stimulation was applied to two electrodes consecutively using the autolab potentiostat. Four shorted stimulation electrodes on the implant served as counter electrodes. In the second experiment, we aimed to mimic the in vivo conditions. We used two different SIROF electrodes as in the experiment before, but on the same implant body. The oven was then set to 37 °C and the experiment with 1 h daily stimulation was run for 60 days. Also in this experiment, the same four shorted stimulation electrodes on the implant served as counter electrodes (Fig. 1a). EIS was recorded in a two-electrode setup with the four shorted stimulation electrodes serving as counter electrode and measured following each daily stimulation. After the termination of the experiment, EIS and CV were recorded in fresh PBS in a three-electrode setup with an Ag/AgCl reference electrode and a stainless-steel counter electrode (~20 cm²). We replicated the 60 days 37 °C experiment with another implant, using one stimulation electrode as WE and two shorted as CE.

### Impedance measures

In vivo electrochemical impedance spectroscopy (EIS) was recorded using the autolab potentiostat (Nova 2.1.6) in a 2-electrode arrangement, with CE located on the implant. EIS was recorded in potentiostatic mode, from 1 Hz to 10 kHz, with an applied RMS voltage of 50 mV overlaid on the open circuit potential (OCP).

### Optical evaluation of electrodes

Implants were pictured using scanning electron microscopy (JEOL 7800 F Prime). Samples were sputtered with 6 nm gold prior. The pictured stimulation electrodes were then classified into categories 1 (like pristine), 2 (light crack formation), 3 (crack formation and delamination <1/3 of electrode area), 4 (delamination > ½ of electrode area), or 5 (heavy delamination).

### Statistical methods

Statistical analyses were performed using GraphPad Prism (v10). For behavioral data, two-sided mixed-effects model ANOVAs were used with restricted maximum likelihood estimation and Geisser-Greenhouse correction to account for repeated measures and missing data. Time was treated as a repeated measure and treatment group as a fixed factor. Tukey's honestly significant difference test was used for post-hoc comparisons, with correction for multiple testing. All tests were two-sided and conducted without data transformation, outlier removal, or normalization. Unpaired two-sided $t$-tests were used to compare impactor parameters, histological measures, and impedance values between stimulation and reserve electrodes. The Shapiro–Wilk test was used to assess normality prior to ANOVA or $t$-tests to determine the need for non-parametric alternatives. Significance was defined as $p \leq 0.05$. Error bars represent the standard error of the mean (SEM), except in Supplementary Figs. 1E–F and 12A–B, where they represent standard deviation (SD). Data in Figs. 1–3, S1, S12 are derived from individual rats (biological replicate), no experimental replication was conducted within the manuscript.

### Reporting summary

Further information on research design is available in the Nature Portfolio Reporting Summary linked to this article.

## Data availability

The data and code used to generate the main and Supplementary Figs. are provided in the source data file. Source data are provided with this paper.

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

## Acknowledgements

This work was supported by funding from the CatWalk Spinal Cord Injury Trust and the Health Research Council of New Zealand (Cure Programme, HRC/Catwalk Partnership 19/895, and B.H. was supported by HRC Sir Charles Hercus Research Fellowship 24/184/A). Funding was also received from the Neurological Foundation of New Zealand (Project Grant 1941). The authors would like to acknowledge Connor Clement (CC) who analyzed the error ladder data. This work was performed in part at the Chalmers Material Analysis Laboratory, CMAL. Support for M.A and L.M. was further contributed by Freiburg Institute for Advanced Studies (FRIAS) and the Chalmers Gender Initiative for Excellence (GENIE). This work was also supported by the Assistant Secretary of Defense for Health Affairs endorsed by the Department of Defense (US $534,258) through the Spinal Cord Injury Research Program (award HT9425-23-1-0492). Opinions, interpretations, conclusions and recommendations are those of the authors and are not necessarily endorsed by the Department of Defense.

## Author contributions

The first authors (B.Har. and L.M.) contributed to this work equally. The last authors (M.A and D.S.) jointly supervised this work. B.Har., B.R., S.O., M.A., D.S. conceived study and obtained funding. L.M. fabricated and characterized implants and conducted FEM. B.Har., S.L., B.Haz., S.M., B.F. performed surgeries and experiments. B.Har., L.M., B.R., S.L., B.Haz. analyzed data. B.Har., L.M., B.R., S.L., B.Haz., S.O., M.A and D.S. interpreted data and discussed results. B.Har. and L.M. wrote the manuscript. All authors commented on and edited the manuscript.

## Funding

## Competing interests

The authors declare no competing interests.
