## [Transparent Peer Review file · Nature Communications]

Daily electric field treatment improves functional outcomes after thoracic contusion spinal cord injury in rats

Corresponding Author: Dr Bruce Harland

Version 0:

Reviewer comments:

Reviewer #1

(Remarks to the Author)

This paper presents a method of subdural daily electrical stimulation to improve motor function after spinal cord injury (SCI). Such approach was designed by placing a thin-film implant with supercapacitive electrodes (SIROF) under dura mater, which could safely and effectively deliver electric field stimulation for thoracic SCI treatment. Subdural electrical stimulation with daily 2Hz electric field therapy promoted functional recovery of SCI rats, enhanced hind limb function and touch sensitivity without inducing a neuroinflammatory response. However, rejection is recommended at this stage, and substantive innovation discussion as well as detailed supporting data are strongly needed before possible recommendation. Some questions are listed as below.

1. In Figure 1E, average BBB score of the treatment group was around 15. The untreated group had a BBB score at around 12, indicating that untreated rats could also regain walking and supporting their own weight with coordinated movements of the front and back legs. Whether it was due to a lack of the blow depth or some other reason, the untreated group was seemed to regain movement. Detailed explanation for this observation is needed.
2. I noticed that the authors used DLC to assess rat behavior, but no data appeared in the article. I suggest the authors to provide photographs of the behavioral recovery VCR and gait images of the rats for evaluation.
3. There is no relevant spinal cord electrophysiological data before and after plant implantation and injury. Please supplement if necessary to make the results more intuitive.
4. In Figure 3, the author wanted to demonstrate the growth of nerve axons after treatment, but compared with the lateral view of spinal cord, the large top view could better identify the growth of nerve axons at the injured edges. Also, in comparisons, authors should always show similar areas and provide control group images, which is also needed in Figure 2.
5. Judging from the extent of damage in Figure 3, the BBB score above 12 in Figure 1 is seemingly difficult to achieve. Data presentation should be representative and mutually corroborating.
6. In supplementary Figure 6, it is necessary to discuss the long-term biosafety of exogenous residues attached to the surface of spinal cord that cannot be removed. Whether the presence of tissue adhesion caused potential adverse reaction (such as local reactive hyperplasia) on spinal cord tissues?
7. Most fluorescence images are not in high resolution, or out of focus. It is difficult to distinguish important specific features. Please increase the image resolution and consider using arrows for better identification.
8. The third and fourth paragraphs of "Introduction" suggested that EF can stimulate axon growth. However, it seemed that no differences of axon-related indexes were observed. More detailed experimental data and discussion are needed.

Reviewer #2

(Remarks to the Author)

In this manuscript, Harland et al. describe a novel subdural implant for delivering electrical stimulation to the spinal cord below the dura mater. The implant is used to administer low-frequency electrical field therapy across a spinal cord lesion, with the goal of promoting regeneration. The authors evaluate functional recovery (motor and sensory), immune response, regeneration, and implant integrity. Based on my expertise, my review focuses on the biological aspects of the study. While there is a potential benefit of the therapy (the authors discussed this well and provide some functional evidence that it might be indeed working), convincing experimental evidence is missing. Furthermore, in my opinion the required control groups and key experiments are missing.

Major concerns:

A central hypothesis of the paper is that subdural implants offer advantages over epidural ones, as stated in the introduction. The authors suggest that subdural electrodes deliver stronger stimulation with less power. However, no comparison between subdural and epidural implants is presented. This weakens their manuscript. Including a group of animals with epidural implants for direct comparison would significantly strengthen the study.

The implant is placed under the dura over spinal segments T13 to L3, with the injury induced while the implant is already in position. This raises some questions: (i) Does the presence of the implant alter the injury mechanics compared to injuries without an implant in place, as explored in Supplementary Figure 1? (ii) Given that the dura mater needs to be left open to connect the implant to an external connector, how do the authors address potential continuous cerebrospinal fluid (CSF) leakage?

In Figure 2a, the appropriate control group—rats with spinal cord injury but no implant—is missing. This makes it difficult to interpret the immune response results.

Does the stimulation result in a motor responses / muscle responses?

If I understood correctly, all rats receive the same stimulation amplitude over time, despite observations that electrodes delaminate from the polyimide and adhere to the spinal cord, likely affecting stimulation efficacy. When this delamination occurs or when the “status” of the stimulation sites degrades is unclear. The authors should assess stimulation effectiveness (by finding the motor thresholds?) across animals and timepoints and ideally normalize the treatment intensity accordingly.

The assessments in the Figure 3a are not sufficient to draw any conclusions about regeneration. Ideally, other anatomical markers should be assessed (labelling for regeneration associated markers), or anatomical tracing should be performed.

Minor comments:

The treated rats exhibit significantly worse recovery during the first week post-injury compared to controls (Figure 1e). The authors should provide an explanation for this.

The authors should explicitly state that the von Frey test was performed when first discussed in the results section (e.g., lines 157–162). Furthermore, the interpretation of the results may confuse readers unfamiliar with the test. The large variability in the no-injury group affects comparisons with injured and treated groups at weeks 4 and 9, and the conclusion that treated rats develop hypersensitivity is not well-supported. Furthermore, the significant difference between the injured treated and untreated groups already at the first timepoint and onward is surprising and requires further discussion. Overall, the results from this test as presented here are inconclusive.

Reviewer #3

(Remarks to the Author)

Dear Authors,

The research topic is relevant and promising. The obtained results can be used for further comparative studies. The described studies were conducted using many modern methodological approaches, indicating the seriousness of the obtained results.

Minor comments and questions:

lines 120-121: Why did you only start 1 hour of treatment on day 7?

lines 126-128: “During the first week after surgery, the treated group showed slower recovery compared to the non-treated group, with a significant difference observed on day 7...”. How can this phenomenon be explained? In essence, these are the same animals - only trauma, without treatment (as mentioned above, the first stimulation was performed only on day 7 after the trauma modelling). Are these data for the first week of treatment and, accordingly, the second week after the trauma?

line 171: Please indicate the area of counting (μm^2 or mm^2) in the spinal cord tissue of Iba-1(+)-cells and GFAP(+)-cells (Fig. 1).

line 200: Please indicate the area of counting (μm^2 or mm^2) in the spinal cord tissue of B-Tub(+)-cells, GAP43(+)-cells and 5HT(+)-cells (Fig. 3).

lines 306-307: "We were unable to explant any other devices intact at ...". Does this mean that with any removal of the stimulator device (especially when you point out the encapsulation of its body), there is a re-damage to the nervous tissue? Is it possible and if so, how to avoid (or minimize) such re-damage?

line 513: Please indicate the mass of the animals.

line 556: Please describe how the contusion modelling in rats was performed (briefly).

lines 566-567: see lines 120-121. It is unclear, please clarify.

line 664: What is the idea behind choosing a cryosection thickness of $\sim 1000 \mu\text{m}$?

Reviewer #4

(Remarks to the Author)

The authors present a work about the use of low-frequency electrical field to promote axonal regeneration of the lesioned nerve. The injection of long stimulation pulses would benefit from the use of electrodes made of capacitive materials, such as the SIROF. The capability to inject high charge inside the safe voltage limits is critical to minimize tissue damage and facilitate the stability of the electrodes over time. The authors try to validate the stability of these electrodes over time and prolonged stimulation, in both in vitro and in vivo, which is essential to provide the therapy over weeks.

Another key point of the study is the strategy of removing the dura to reduce the electric field strength and deliver a stronger stimulation while using an order of magnitude less power. This approach has the advantage of reducing the current level, which is critical for the safe injection of long pulses, such as the one required for electric field treatments

In summary, it is recognized the value of exploring the use of high-performing electrodes, in combination with a different implantation strategy, however, the conclusions are not always supported by comprehensive data and statistical significance.

In Figure 5, the authors show the electrochemical characterization of one SIROF electrode, to prove stability after the aging test, and in combination with electrical stimulation. The plots presented show good stability for the impedance and cyclic voltammetry of one electrode, however, statistics on a larger number of electrodes are required to ensure the significance of the results.

In the paper, the use of capacitive SIROF electrodes is presented as an advantage to inject long pulses of 250 ms in the safe voltage window. The electrochemical characterization should include the measure of the voltage polarization of the stimulation pulses used in this study to evaluate if the voltage shift at the electrode-tissue interface is inside the voltage-safe limits.

In Figure 6a, the impedance spectrum of the electrodes in vivo is used to ensure the functionality of the electrodes over time. The fact that the magnitude is below the open circuit potential limit, is it enough to claim that the electrodes are functional? Characteristics such as the shape of the impedance spectrum need to be considered. The presence of a 'noisy measurement' for the impedance magnitude of some electrodes, as well as the change of the phase, are indications that the electrodes cannot function properly, at least in the in vivo settings. How is the measurement of the cyclic voltammetry over weeks of implantation?

Since the in vivo functionality of the electrodes in vivo, is intended in the perspective of delivering the therapeutic treatment over time, the voltage polarization of the stimulation pulses should be included in the characterization. After the increase of the impedance over weeks of implantation, can the electrodes still inject the pulses inside the safe voltage limits to avoid faradaic reactions that can damage both electrodes and tissue? Can the voltage compliance of the stimulator, deliver the voltage needed to stimulate even after several weeks?

The presence of tissue encapsulation around the device, produced by the foreign body reaction (FBR) after weeks of implantation, raises a question in relation to the advantage of removing the dura when the treatment is extended over time. to improve the penetration of the electric field and the efficacy of the treatment. In the text it is cited a study (doi: 10.1063/5.0163264) to support the statement that the presence of FBR, has little effect on the field distribution in the white and gray matter. However, in this model, the FBR is considered in a device implanted on top of the dura. In the case of subdural electrodes, how is it expected to change the electric field after weeks of implantation? The advantage of having electrodes below the dura to improve the penetration of the electric field and the efficacy of the treatment is still an advantage after weeks?

Has been any test or analysis performed to evaluate that not only the electrodes but also the polyamide encapsulation of the device is stable over 12 weeks of in vivo implantation, to exclude liquid permeation inside the polymeric sandwich which can produce the electrodes short-cut and consequent change in the stimulation current spreading?

The authors use 4 electrodes shorted as a counter path for the stimulation current. It is visible in the design a big rectangular electrode, which I suppose was designed in the first place, to work as a counter. Could the authors comment on the choice of using 4 electrodes shorted, instead of the bigger rectangular electrode?

Version 1:

Reviewer comments:

Reviewer #2

(Remarks to the Author)

The authors have addressed some of my comments very thoroughly, providing additional data in Figure 4 and Supplementary Figure 12. However, for certain points, they have “only” acknowledged the study’s limitations, discussing them further in the result section and suggesting including them in future studies.

While this is somewhat disappointing, fully addressing these concerns would require repeating substantial parts of the study to provide new data on testign motor responses and adjusting the stimulation amplitudes, or performing CNS tract tracing to properly evaluate regeneration. Given that these would be very extensive experiments and the fact that one of the manuscript’s key innovation is the development and safety assesment of the subdural implant, their response may be acceptable, provided that the other reviewers and editors agree on the study’s sufficient novelty.

Reviewer #4

(Remarks to the Author)

The revised manuscript addresses the majority of the reviewers’ comments. The inclusion of additional experiments, such as long-term stimulation assessments, in vitro voltage transient measurements, and FIB analysis for evaluating the adhesion of the PI layers, is appreciated and strengthens the overall conclusions of the paper. Despite certain limitations, particularly regarding the long-term in vivo evaluation of electrode functionality, as a first proof-of-concept study, the authors provide sufficient evidence to support the work.

The images or other third party material in this Peer Review File are included in the article’s Creative Commons license, unless indicated otherwise in a credit line to the material. If material is not included in the article’s Creative Commons license and your intended use is not permitted by statutory regulation or exceeds the permitted use, you will need to obtain permission directly from the copyright holder.

REVIEWER COMMENTS

Reviewer #1 (Remarks to the Author):

This paper presents a method of subdural daily electrical stimulation to improve motor function after spinal cord injury (SCI). Such approach was designed by placing a thin-film implant with supercapacitive electrodes (SIROF) under dura mater, which could safely and effectively deliver electric field stimulation for thoracic SCI treatment. Subdural electrical stimulation with daily 2Hz electric field therapy promoted functional recovery of SCI rats, enhanced hind limb function and touch sensitivity without inducing a neuroinflammatory response. However, rejection is recommended at this stage, and substantive innovation discussion as well as detailed supporting data are strongly needed before possible recommendation. Some questions are listed as below.

1. In Figure 1E, average BBB score of the treatment group was around 15. The untreated group had a BBB score at around 12, indicating that untreated rats could also regain walking and supporting their own weight with coordinated movements of the front and back legs. Whether it was due to a lack of the blow depth or some other reason, the untreated group was seemed to regain movement. Detailed explanation for this observation is needed.

The contusion injury model we used is widely recognized in the field, and even with severe injury forces, rats consistently exhibit some degree of hind-limb functional recovery. We purposely selected an injury severity of 175 kDynes (impact force) to achieve an intermediate level of recovery in the untreated group (~12 out of 21 on the BBB score). This level of recovery allows sufficient room for the applied treatment to either reduce or improve motor function, providing meaningful insight into its effects. The observed recovery in the untreated group was intentional and not due to insufficient impact depth or any other methodological issue. It aligns closely with recovery levels in controls from our previous work using the same impact force and impactor device, centered at T10. This comparison is provided below to demonstrate that the untreated control group had the level of injury that we intended (Fig. R1).

Figure R1: Comparison of BBB scores for non-treated group in the current study (injury + implant) vs an injury only group from Griffin et al., 2020, which received the same injury severity (175 kDynes) using the same impactor device centred at spinal process T10.

We chose not to employ a more severe injury for the following reasons:

1. **Task feasibility:** Animals required some degree of weight-bearing for the Error Ladder task, and plantar foot placement was essential for the von Frey task. A more severe injury may have negatively affected data collection in these tasks.
2. **Health and welfare considerations:** We aimed for a long recovery period (12 weeks) to thoroughly observe treatment-induced changes. Animals with more severe injuries or complete loss of function are significantly more prone to health and welfare complications over extended recovery periods.
3. **Relevance to translational research:** A more severe injury might represent overly challenging experimental conditions, where a treatment that works for moderate injuries could appear ineffective. For translation to human applications, therapeutic strategies are likely to first target moderately injured patients before advancing to those with more severe injuries.

We have included a summary of this reasoning in the method section, and in the caption of Supplementary Fig. 1.

2. I noticed that the authors used DLC to assess rat behavior, but no data appeared in the article. I suggest the authors to provide photographs of the behavioral recovery VCR and gait images of the rats for evaluation.

We did not use DeepLabCut (DLC) to assess rat behaviour. Instead, DLC was utilized solely to blind investigators when analysing the BBB videos by detecting the implant (external "Backpack") and blurring the painted flag on the back, which was used to identify individual animals. This step was undertaken to ensure the videos were blinded before being evaluated using the BBB scale. Additionally, all file names were anonymized to maintain blinding during the assessment process.

The supplementary video included in our submission, which demonstrates the locomotor function of five treated and untreated rats at weeks 11/12, does not include DLC-based blinding. This is because the video was created using the original, unblinded, and uncompressed footage. The blurred backpacks processed via DLC is demonstrated below in Fig. R2.

Figure R2: Deep Lab Cuts software was used to blur out the identifying painted flag on each implant to facilitate blinded assessment of the BBB analysis. Still image from BBB video is shown (A) before, and (B) after deep labs cuts is applied.

We are interested in exploring the potential use of DLC in the future to detect limb positions and generate gait images, and we are aware of other research groups investigating similar applications. However, implementing this in an open-field environment poses several challenges. Achieving accurate results would require multiple cameras positioned at various angles, and even with this setup, accurately labelling leg kinematics in rats could be problematic. Fur often obscures critical landmarks such as the knee and hip joints, which might necessitate shaving the hind region or attaching sensors to improve joint visibility. However, in injured animals, this approach could exacerbate existing welfare concerns, as these animals already face challenges with temperature regulation. Removing fur may further compromise their ability to maintain body temperature, raising significant ethical considerations.

3. There is no relevant spinal cord electrophysiological data before and after plant implantation and injury. Please supplement if necessary to make the results more intuitive.

We are uncertain about the exact suggestion being made as before the implant is in place there are no electrodes we can record from. Collecting electrophysiology data from the spinal cord prior to implantation would require an evoked potential method involving invasive needle electrodes inserted into target leg muscles. Additionally, a second electrode would need to be applied to the surface of the spinal cord or inserted into it. Implementing these steps would significantly add to the length and complexity of an already demanding surgical procedure. If the reviewer is instead suggesting spinal recordings in freely moving rats before and after injury and treatment, this would require two major surgeries. The first would involve implanting the device, closing the skin, and allowing the animal to recover before performing electrophysiology. The second surgery would reopen the site to apply the injury. Such an approach would significantly increase the complexity of the experiments and could negatively affect the animals' welfare and recovery.

While we acknowledge the value of electrophysiological data in certain contexts, we are unclear how such data would enhance the intuitiveness or impact of our current findings. Our study focuses on functional and behavioural outcomes of the electric field treatment, which are the most directly relevant measures for assessing its efficacy.

4. In Figure 3, the author wanted to demonstrate the growth of nerve axons after treatment, but compared with the lateral view of spinal cord, the large top view could better identify the growth of nerve axons at the injured edges. Also, in comparisons, authors should always show similar areas and provide control group images, which is also needed in Figure 2.

We chose to cut sagittal longitudinal sections to provide a lateral view of the spinal cord and lesion, as we were specifically interested in analysing the tissue dorsal to the lesion (between the lesion and electrodes), where the electric field was strongest. These sections capture the dorsal tissue of interest as well as the ventral tissue, allowing us to investigate potential differences between these regions.

While we considered cutting frontal longitudinal sections (from dorsal to ventral), we opted against this approach. During longitudinal cutting, the outermost sections, where the tissue is thinnest, are particularly prone to tearing, curling, or deformation, especially in the presence of a lesion. For frontal sections, this would have jeopardized the integrity of the critical region between the lesion and electrodes.

Following the reviewer's suggestion, we have updated Figures 2 and 3 to include examples from both treated and non-treated groups, ensuring they highlight similar tissue areas for direct comparison.

5. Judging from the extent of damage in Figure 3, the BBB score above 12 in Figure 1 is seemingly difficult to achieve. Data presentation should be representative and mutually corroborating.

The images in Fig. 3 show the lesion epicenter, where the lesion is widest in both the dorsal-ventral and rostral-caudal aspects, spared tissue at the lateral of the lesion cannot be visualised in these sections. The finding that rats with this extent of damage can recover weight bearing plantar stepping is consistent with other studies using a similar injury model in rats. For example, in Griffin et al. (2020) we utilized the same impactor device and force, reporting comparable BBB recovery in the control group (as referenced in our response to Reviewer Comment 1). Coronal sections of the lesion epicenter from that study also demonstrate a comparable lesion size.

[REDACTED]

Figure R3: Figure from Griffin et al. 2020 using the same impactor device and force but aimed at T10 using the same strain/gender/age rats. (a-d) Show representative Luxol fast blue-eosin staining of the lesion epicenter in four experimental conditions: (a) contusion alone, (b) contusion with AAV-ADAMTS4 treatment, (c) contusion with rehabilitation, and (d) contusion with both AAV-ADAMTS4 treatment and rehabilitation.

Furthermore, another study by Ward et al. (2014) using the same impactor, albeit with a slightly higher force (210 Kilodyne) directed at T9 in rats, also demonstrates comparable tissue loss at the lesion epicenter, with comparable BBB scores (<https://doi.org/10.1089/neu.2013.3082>, Figure 9 on page 829). To improve clarity, we have added a sentence and these references to the start of the histology results section in the manuscript.

While we maintain that the extent of damage shown at the lesion epicenter, as well as the functional recovery of the rats, is consistent with previously published studies, we acknowledge the reviewer's point that additional comparisons of lesion size and functional recovery would provide greater context for the reader. To address this:

1. We have included more examples of histology from different regions of the lesion in a new Supplementary Fig. 11 to show different cross-sections through the lesion.
2. In Fig. 3, we have clarified in the caption that these images are specifically from the lesion epicenter.

These additions aim to strengthen the interpretation of our results and overall clarity of the lesion extent.

References: Griffin JM, Fackelmeier B, Clemett CA, Fong DM, Mouravlev A, Young D, O'Carroll SJ. (2020). Astrocyte-selective AAV-ADAMTS4 gene therapy combined with hindlimb rehabilitation promotes functional recovery after spinal cord injury. *Exp Neurol.*, 327:113232. <https://doi.org/10.1016/j.expneurol.2020.113232>

Ward P, Herrity AN, Smith RR, Willhite A, Harrison BJ, Petruska JC, Harkema SJ, Hubscher CH. (2014). Novel Multi-System Functional Gains via Task Specific Training in Spinal Cord Injured Male Rats. *Journal of Neurotrauma*, 31:9, 819-833. <https://doi.org/10.1089/neu.2013.3082>.

6. In supplementary Figure 6, it is necessary to discuss the long-term biosafety of exogenous residues attached to the surface of spinal cord that cannot be removed. Whether the presence of tissue adhesion caused potential adverse reaction (such as local reactive hyperplasia) on spinal cord tissues?

The residues mentioned by the reviewer originate from the delamination of the electrode surface, either due to mechanical slipping or the explantation process. The observed electrode delamination was unintended. It has to be noted that in Supplementary Fig. 6, the stimulation electrode shown in panel A is situated below the dura mater on the spinal cord surface. However, the large ground electrode shown in panel B is outside the dura mater and spinal cord; we have now made this clear in the figure caption text. The residues consist of Pt and SIROF, both are well-established, biocompatible materials with no reported adverse reactivity *in vivo*. Cochlear implants are thoroughly investigated as they are implanted in humans since the 1970s. Indeed, it has been reported that Pt was found in the surrounding sheath of cochlear implants. However, in the majority of cases, these Pt residues do not impact device performance, nor cause issues in biosafety, making electrode residuals an acceptable risk (Nadol Jr. et al. 2014).

In this study, the extent of electrode delamination is expected to be comparably small, as the electrodes consist of thin-films and most of the electrode's material remained attached to the substrate (Fig. 7B, Supplementary Fig. 7). Furthermore, the extent of electrode delamination *in vivo* remains uncertain as we could only perform an end-point analysis making it impossible to distinguish between delamination during the treatment period from delamination resulting from the explantation procedure. Nonetheless, the residues are expected to be minor in quantity and composed of low reactivity materials. Consequently, we do not expect any significant influence of these residues on the applied electric field treatment. This aspect has been highlighted in the discussion section (l. 468).

The primary objective of this study was to evaluate the feasibility of subdural electric field delivery and its potential for functional improvement following SCI. Our intention in showing this Supplementary Fig. 7 was to alert other researchers of the possibility of delamination when larger metallic surfaces are used. In a future study we have incorporated strategies to mitigate delamination, such as hydrogel coating to reduce adhesion forces on the metal surface, and the use of multiple interconnected small electrodes instead of a single large electrode (Fig. R4). In the future we plan to also incorporate adhesion promoting layers. We have submitted a study demonstrating the enhanced delamination resistance of small, interconnected electrodes to the 47th Annual International Conference of the IEEE Engineering in Medicine and Biology Society (EMBC 2025).

Figure R4. Small, interconnected electrodes instead of a large electrode to enhance the resistance to delamination.

Reference: Nadol JB Jr, O'Malley JT, Burgess BJ, Galler D. (2014) Cellular immunologic responses to cochlear implantation in the human. *Hear Res.* 318:11-7. <https://doi.org/10.1016/j.heares.2014.09.007>

7. Most fluorescence images are not in high resolution, or out of focus. It is difficult to distinguish important specific features. Please increase the image resolution and consider using arrows for better identification.

The histology images in the original submission were unintentionally affected by file compression. This has now been corrected for all histology images in the new version. We have provided additional images to show examples from each of the groups and improved the annotation.

8. The third and fourth paragraphs of "Introduction" suggested that EF can stimulate axon growth. However, it seemed that no differences of axon-related indexes were observed. More detailed experimental data and discussion are needed.

We appreciate the reviewer's insight, our study reports in the last paragraph of the introduction, "*We did not observe significant differences in markers related to axon density and axonal regeneration around the injury site.*" We have now bolstered the discussion in the revised paper to address the following points:

- 1) The presented data may not indicate a lack of regeneration, but rather that it may have been undetectable with our injury model and assessment methods. Our chosen markers (GAP43 and β III-tubulin) were selected based on their proven efficacy in detecting axonal growth differences in epidural EF treatment studies (Bacova et al., 2019, 2022). However, while Bacova et al. assessed axonal markers at 4 and 8 weeks, our 12-week endpoint may have surpassed the optimal window for detecting early regenerative changes, potentially contributing to the lack of measurable differences.
- 2) Other regeneration-associated markers (e.g., neurofilament, BDNF) also present challenges in long-term studies. Neurofilaments indicate axonal integrity but not regeneration, and BDNF expression, like GAP43, is transient and may have declined by 12 weeks. To improve detection of axonal regeneration, we propose using viral tracers (e.g., GFP-expressing AAV) in future studies to label corticospinal or rubrospinal tracts, offering direct evidence of axon extension and connectivity, independent of regeneration timing.

- 3) A critical distinction is that Bacova et al. used a compression injury model, producing a focal lesion with minimal tissue loss, whereas our contusion injury model generates a larger cystic cavity, presenting greater challenges for detecting localized axonal changes. The contusion model, widely used for its relevance to clinical SCI, replicates glial scar formation and pathophysiological progression, enhancing translational applicability. Despite the strengths of the contusion model, a limitation is histological changes in tissue are harder to detect.
- 4) Assessing 3D axon growth using thin-section histology in the context of a large lesion cavity presents inherent limitations. In future studies we plan to address this by integrating viral tracers with volumetric imaging of thicker sections or whole spinal cords, utilizing confocal and light-sheet microscopy.
- 5) Another possibility is that the electric field is promoting functional recovery through mechanisms other than axonal regeneration. The research field is yet to determine which mechanisms contribute the largest to the recovery of function, for example, others have shown that EF treatment in animals reduced the amount of reactive astrocytes and reduced demyelination, both factors could promote functional recovery but wouldn't have been picked up by our assessment methods.

When seeking mechanisms which drive functional recovery, it is only possible to look at a certain number of markers in a finite number of animals. We agree with the reviewer that the study would benefit from a more detailed discussion, which we now have added. To support advancement in this exciting field we put our focus on assisting the rational design of experiments in future work.

Reference: Bacova M, Bimbova K, Fedorova J, Lukacova N, Galik J. (2019). Epidural oscillating field stimulation as an effective therapeutic approach in combination therapy for spinal cord injury. *J Neurosci Methods*. 311:102-110. <https://doi.org/10.1016/j.jneumeth.2018.10.020>

Bacova M, Bimbova K, Kisucka A, Lukacova N, Galik J. (2022). Epidural oscillating field stimulation increases axonal regenerative capacity and myelination after spinal cord trauma. *Neural Regen Res*. 17(12):2730-2736. <https://doi.org/10.4103/1673-5374.339497>

Reviewer #2 (Remarks to the Author):

In this manuscript, Harland et al. describe a novel subdural implant for delivering electrical stimulation to the spinal cord below the dura mater. The implant is used to administer low-frequency electrical field therapy across a spinal cord lesion, with the goal of promoting regeneration. The authors evaluate functional recovery (motor and sensory), immune response, regeneration, and implant integrity. Based on my expertise, my review focuses on the biological aspects of the study. While there is a potential benefit of the therapy (the authors discussed this well and provide some functional evidence that it might be indeed working), convincing experimental evidence is missing. Furthermore, in my opinion the required control groups and key experiments are missing.

Major concerns:

A central hypothesis of the paper is that subdural implants offer advantages over epidural ones, as stated in the introduction. The authors suggest that subdural electrodes deliver stronger stimulation with less power. However, no comparison between subdural and epidural implants is presented. This weakens their manuscript. Including a group of animals with epidural implants for direct comparison would significantly strengthen the study.

We thank the reviewer for this insightful comment and agree that including a group of animals with epidural implants for direct comparison would enhance the study. However, implementing epidural stimulation in a comparable manner would require substantial efforts, including adaptations to the

implant design, surgery, and injury model, as well as the inclusion of additional control groups (e.g., epidural implant without injury or stimulation). Unfortunately, conducting these experiments within a reasonable timeframe is not feasible for the current study. Our expectation formulated in the introduction that subdural implants need less power than epidural implants is supported by literature. In the submitted manuscript, we cited Huang et al., whose simulations of the human spinal cord demonstrated that epidural stimulation requires an order of magnitude larger current amplitude than subdural stimulation to recruit axons in the dorsal column. Their nerve fibre model utilised the voltage distribution within the spinal cord, effectively showing that achieving a comparable electric field (EF) inside the cord with epidural stimulation would require significantly higher current amplitudes compared to the subdural approach.

To further support this hypothesis, we have included additional clinical evidence from Howell et al. In their study, the same percutaneous array was positioned both subdurally and epidurally in patients to assess stimulation efficiency. Sensory thresholds ranged from 0.2 to 0.8 mA for subdural placement, compared to 1.3 to 19 mA for epidural placement. Using these clinical findings, the authors developed a patient-specific model to evaluate axon recruitment and concluded that “Intradural placement dramatically increased stimulation efficiency and reduced the power required to stimulate the dorsal columns by more than 90%.”

In addition to incorporating this literature, we have added a comment about the absence of a group with epidural stimulation to the limitation section in the discussion. Furthermore, we utilized the model of the rat’s spinal cord and shifted the implant epidurally. We found that the average longitudinal EF strength in the lesion for the epidural implant is 13% less than for the subdural implant. We included these findings in Fig. 4 and in the accompanying section. While 13% may seem modest, especially in light of referenced statements, the anatomical differences between humans and rats must be considered. For an average human, the thickness of dura is 0.3 mm, the distance between dorsal surface and dura is 3.2 mm, and the diameter of the spinal cord is 8 mm (Zander et al. 2020). This means that in a human a subdural implant needs to be shifted at least 3.5 mm dorsally to be epidural. In contrast, for a rat with a 3 mm cord diameter, the shift would be just 0.125 mm. Given that EF strength decreases quadratically with distance (assuming that the electrodes are point charges), this difference in displacement significantly impacts the EF strength. Despite pointing this out, we are not necessarily advocating that a subdural approach will be best in larger animal model as many more things will also change, such as the size of the implant. To date, most of the work focuses on epidural stimulation. It thus requires more work to assess the efficacy of subdural stimulation. In this rat experiment our aim was to provide the best situation for the EF to allow the most focussed and penetrative EF possible, which is provided by the subdural approach. We added these aspects to the discussion section (l. 365).

Reference: Zander HJ, Graham RD, Anaya CJ, Lempka SF. (2020). Anatomical and technical factors affecting the neural response to epidural spinal cord stimulation. *Journal of neural engineering*. 17;17(3):036019. <https://doi.org/10.1088/1741-2552/ab8fc4>

The implant is placed under the dura over spinal segments T13 to L3, with the injury induced while the implant is already in position. This raises some questions: (i) Does the presence of the implant alter the injury mechanics compared to injuries without an implant in place, as explored in Supplementary Figure 1? (ii) Given that the dura mater needs to be left open to connect the implant to an external connector, how do the authors address potential continuous cerebrospinal fluid (CSF) leakage?

In Figure 2a, the appropriate control group—rats with spinal cord injury but no implant—is missing. This makes it difficult to interpret the immune response results.

Our approach of inserting the thin film implant subdurally before inducing the injury enhances the ease of implantation and we believe leads to a more consistent surgical procedure. This pre-injury placement prevents complications arising from injury-induced debris and bleeding which can be variable across impacts injuries. Importantly, treated and non-treated groups received the impaction with implants already in place, thus in identical procedures. The graphs in Supplementary Fig. 1B, C demonstrate that similar force vs time profiles of the impaction were achieved for the treated and non-treated groups. Furthermore, prior insertion reduces the need for multiple surgeries which has welfare implications, can negatively affect recovery, and impact possible treatment effectiveness. Nevertheless, we agree that it would strengthen the manuscript to provide a comparison of the injury mechanics with and without the implant in place.

To this end, we include an additional Supplementary Fig. 12, comparing the impact profile of implanted rats in the current study (combined treated and non-treated groups) with rats from our previously published study (Meissner et al. 2024). In this previous study, all rats had the same laminectomy (removal of processes T10-T12) and then received the same 175 Kilodyne impact at the same position (below spinal process T11). Therefore, the injury mechanics can be directly compared. We found that performing implant insertion followed by delivering the spinal cord impact directly on top of the implant within the same surgery does not alter the injury's impact mechanics. We added a comment about this comparison in the methods section (l. 606). The rats in Meissner et al. then received various subdural infusions of hydrogels and drugs after the impact injury, so we have not attempted to compare the animals post-injury. We also include comparison of pre- and - post impaction impedance measurements from no-injury vs injury rats in the current study, to provide data to show the impact delivered over the thin-film implant does not alter function of the electrodes and tracts. We hope that this data will help to establish this method as a useful technique for others.

Supplementary Figure 12: Performing implant insertion followed by delivering the spinal cord impact directly on top of the implant within the same surgery does not alter the injury's impact mechanics or affect electrode impedance. This approach avoids potential damage from sliding the implant in after the impact when swelling and bleeding are present, and eliminates the need for multiple surgeries that could affect animal welfare and recovery. **(A)** Comparison of impact max impact force (kDynes) and **(B)** force over time mechanics between rats with a subdurally positioned implant and those without an implant. Data include the implanted rats from this study (combined Treated and non-treated groups) compared with animals from our previous study (Meissner et al., 2024) that underwent the same laminectomy and impact force / location, in the absence of an implant. **(C)** Delivering the impact on top of the thin film implant did not affect impedance measurements from the electrodes; we compare pre- and -post impactation impedance measurements from no-injury vs injury rats in the current study.

In this study, cerebrospinal fluid leakage was mitigated by applying Gelfoam as a hemostatic seal over the exposed spinal cord in the space left by the removed spinal processes. This method seems to work well in rats as we did not observe any motor impairments or other welfare issues in the no-injury group, which had the implant and Gelfoam. However, we acknowledge that a different approach would be necessary in larger mammals, including humans and now state this in the limitations section of the discussion.

We did not include a group in this manuscript that received an injury but no implant. The focus of this work was functional and behavioural outcomes of the electric field treatment for which the non-treated group that also had the implant was the most meaningful control. Before this study the immune response to subdural electric field treatment was unknown. It was therefore crucial to investigate its effect. We show in Fig. 2A that the treatment does not elicit an additional immune response compared to controls with the same implant and injury but without electric field treatment. With a control group with no implant and the same injury we would not have been able to draw this conclusion as differences in immune reaction could be attributed to the presence of the implant or the electric field treatment. To gauge the immune response of the implant itself, we have previously compared implanted vs non-implanted rats in the absence of an injury (Harland et al., 2022) and found no significant differences in Iba1 and GFAP expression. We now reference to our previous work in the result section to further clarify the choice of experimental groups.

References: Meissner S, Lopez S, Rees S, O'Carroll S, Barker D, Harland B, Raos B, Svirskis D. (2024). Safe subdural administration and retention of a neurotrophin-3-delivering hydrogel in a rat model of spinal cord injury. *Sci Rep.* 14(1):25424. <https://doi.org/10.1038/s41598-024-77423-5>

Harland B, Aqrave Z, Vomero M, Boehler C, Cheah E, Raos B, Asplund M, O'Carroll SJ, Svirskis D. (2022). A Subdural Bioelectronic Implant to Record Electrical Activity from the Spinal Cord in Freely Moving Rats. *Adv Sci.* 9(20):e2105913. <https://doi.org/10.1002/advs.202105913>

Does the stimulation result in a motor responses / muscle responses?

The electric field stimulation used in our study is intentionally set well below the threshold to elicit motor or muscle responses. Generally, longer pulse widths are less likely to evoke direct muscle contractions compared to shorter pulse widths at the same amplitude. Studies in neuromodulation, such as those by the Courtine group (Kathe et al. 2022, Capogrosso et al., 2018), commonly utilize stimulation parameters of 0.5–5 mA amplitudes, 200–400 μ s pulse widths, and frequencies of 20–200 Hz, which have been shown to remain subthreshold for generating visible motor responses or muscle twitches.

In contrast, our stimulation uses significantly lower current (5 μ A) and much longer pulse widths (250 ms), resulting in a stimulation frequency of only 2 Hz. However, due to the fact that our electrodes are placed directly on the surface of the spinal cord, we carefully monitor each animal during initial stimulation sessions to ensure there are no unintended motor responses. We added this explanation to the method section.

References: Kathe C, Skinnider MA, Hutson TH. *et al.* The neurons that restore walking after paralysis. *Nature* **611**, 540–547 (2022). <https://doi.org/10.1038/s41586-022-05385-7>

Capogrosso M, Wagner FB, Gandar J. *et al.* Configuration of electrical spinal cord stimulation through real-time processing of gait kinematics. *Nat Protoc* **13**, 2031–2061 (2018). <https://doi.org/10.1038/s41596-018-0030-9>

If I understood correctly, all rats receive the same stimulation amplitude over time, despite observations that electrodes delaminate from the polyimide and adhere to the spinal cord, likely affecting stimulation efficacy. When this delamination occurs or when the “status” of the stimulation sites degrades is unclear. The authors should assess stimulation effectiveness (by finding the motor thresholds?) across animals and timepoints and ideally normalize the treatment intensity accordingly.

This is correct, each rat receives the same stimulation amplitude over time. While we model the electric field to estimate its strength and penetration, we cannot be certain that it remained

consistent over the 12 weeks, as increased impedance may indicate electrode changes or encapsulating tissue buildup.

At the end of the experiment, we observed some electrode delamination, though only one in ten showed delamination that would render it non-functional. Additionally, the larger electrode surfaces sometimes adhered to the tissue, making it unclear whether delamination occurred *in vivo* or during dissection.

We appreciate this insightful suggestion. Establishing baseline stimulation parameters to evoke motor function and tracking changes in this threshold throughout the experiment would provide valuable information. We will consider this for future studies as it could provide another avenue to probe electrode function over time and added this aspect in the discussion section (line 462).

The assessments in the Figure 3a are not sufficient to draw any conclusions about regeneration. Ideally, other anatomical markers should be assessed (labelling for regeneration associated markers), or anatomical tracing should be performed.

Our approach was to use GAP-43, a regeneration-associated protein that indicates axonal sprouting and plasticity, making it a direct marker of regeneration. We co-stained with β III-tubulin as a control for overall neuronal presence, helping to contextualize the results. This approach was selected in part because it has been shown to effectively highlight differences in epidural electric field treatment versus SCI controls in several studies (Bacova et al., 2019, 2022). These studies used a combination of GAP-43 and neurofilament. It is important to note that a compression injury model was used, which produces a more focal lesion with minimal cavitation, as evident in their histological images. In contrast, our contusion injury model results in a much larger cystic cavity and loss of tissue. The markers we have chosen don't show significant differences within the longitudinal sagittal sectioning method we have employed, but locomotion improved. Ultimately, regaining function is the main goal.

While GAP-43 is a widely recognized marker for axonal regeneration, it is known to have a transient expression pattern, with levels peaking during early stages of regeneration and declining in later phases. Since our histological analysis was conducted 12 weeks post-injury, it is possible that we missed the window of strong GAP-43 expression. This limitation could explain the relatively low levels of GAP-43 observed in our study. Other commonly used regeneration associated markers, such as neurofilament and BDNF, also present challenges when used in long-term studies. Neurofilaments, while indicative of axonal integrity, do not specifically reflect regeneration, and BDNF expression is similarly transient and may have declined by the 12-week time point.

To overcome these limitations in future studies, we propose the use of viral tracer injections in the brain; a self-complementary AAV encoding a fluorescent reporter (*e.g.*, GFP), injected into the sensorimotor cortex in order to label axons in the corticothalamic tract, or into the red nucleus to label the rubrospinal tract. Unlike regeneration-associated markers, axonal tracers are independent of the timeline of regeneration and can provide direct evidence of axonal extension and connectivity.

We have summarised this response and added it to a limitations section at the end of the discussion.

Reference: Bacova M, Bimbova K, Fedorova J, Lukacova N, Galik J. (2019). Epidural oscillating field stimulation as an effective therapeutic approach in combination therapy for spinal cord injury. *J Neurosci Methods*. 311:102-110. <https://doi.org/10.1016/j.jneumeth.2018.10.020>

Bacova M, Bimbova K, Kisucka A, Lukacova N, Galik J. (2022). Epidural oscillating field stimulation increases axonal regenerative capacity and myelination after spinal cord trauma. *Neural Regen Res*. 17(12):2730-2736. <https://doi.org/10.4103/1673-5374.339497>

Minor comments:

The treated rats exhibit significantly worse recovery during the first week post-injury compared to controls (Figure 1e). The authors should provide an explanation for this.

We have considered this question ourselves, and our response remains speculative. We hypothesize that the mechanical forces during the physical connection of the rats' backpack to the stimulator may have caused additional tissue damage. While we carefully manage this process to minimize strain, and the lightweight overhead cable is kept loose, tethering forces can still exert an impact. Indeed, in Vomero et al. (2022), tethered devices implanted in the rat's brain had enhanced gliosis, increased reactive astrocyte encapsulation and reduced tissue integration compared to untethered devices. Notably, however, the tethered devices in their study were plugged in for shorter durations and less frequently compared to ours, which may further amplify the effects. In our study, animals in the control and untreated group were not physically connected to the stimulator. We have included this explanation and reference in the discussion.

If our hypothesis is correct, the treated animals had a more severe injury but still recovered better. This further emphasizes the efficacy of electric field treatment. Looking ahead, we envision a fully integrated system that eliminates the need for repeated physical connections, thereby minimizing potential mechanical stress.

Reference: Vomero M, Ciarpella F, Zucchini E, Kirsch M, Fadiga L, Stieglitz T, Asplund M. On the longevity of flexible neural interfaces: Establishing biostability of polyimide-based intracortical implants. (2022). *Biomaterials*. 281:121372. <https://doi.org/10.1016/j.biomaterials.2022.121372>

The authors should explicitly state that the von Frey test was performed when first discussed in the results section (e.g., lines 157–162). Furthermore, the interpretation of the results may confuse readers unfamiliar with the test. The large variability in the no-injury group affects comparisons with injured and treated groups at weeks 4 and 9, and the conclusion that treated rats develop hypersensitivity is not well-supported. Furthermore, the significant difference between the injured treated and untreated groups already at the first timepoint and onward is surprising and requires further discussion. Overall, the results from this test as presented here are inconclusive.

We have modified the first sentence of the von Frey results paragraph to state that an electronic von Frey task was used.

We acknowledge that the von Frey test is inherently variable, particularly in the no-injury or sham groups, as highlighted in prior work (Cobos & Portillo-Salido, 2013). This variability is an important factor in interpreting the week 4 and week 9 differences observed between treated and no-injury groups. We have now addressed this point in the results and included the relevant reference in the text for clarity.

Despite its inherent variability, the von Frey test is widely validated as a reliable measure of mechanical sensitivity. The consistent differences observed between treated and non-treated groups over the 12-week period provide strong evidence of improved touch sensitivity in the treated animals. However, we recognize that the difference observed at week 1 is unexpected. While we had not previously commented on this, as its underlying cause remains uncertain, we acknowledge that it warrants further discussion.

This difference is primarily driven by a quicker average withdrawal in the treated group compared to any later timepoint. It may reflect an early effect of the electric field treatment on the acute stage of injury, a period marked by a profound cascade of biological and molecular changes that differ significantly from later phases of recovery. For instance, the electric field may have heightened the

excitability of nociceptive pathways, modulated inflammatory processes, or disrupted acute pain signalling more effectively, potentially resulting in lower von Frey thresholds during this phase.

Alternatively, as noted in the methods, some measures in injured rats could not be taken during the first weeks due to non-plantar paw positioning, which was particularly pronounced during week 1. Consequently, the sample size for this timepoint is reduced, which may contribute to the observed variability and the statistical significance at this early stage. These factors highlight the need for cautious interpretation of the week 1 data, therefore we have added brief discussion of these points in the new limitations section.

Reference: Cobos, E.J., & Portillo-Salido, E. (2013). "Bedside-to-Bench" Behavioral Outcomes in Animal Models of Pain: Beyond the Evaluation of Reflexes. *Current Neuropharmacology*. 11(6):560-91. <https://doi.org/10.2174/1570159x113119990041>

Reviewer #3 (Remarks to the Author):

Dear Authors,

The research topic is relevant and promising. The obtained results can be used for further comparative studies. The described studies were conducted using many modern methodological approaches, indicating the seriousness of the obtained results.

Minor comments and questions:

lines 120-121: Why did you only start 1 hour of treatment on day 7?

Surgeries were all conducted Monday to Friday, and for all rats 1 hour daily treatment started the day after the implant/injury surgery, day 1. Then, every treated rat received daily treatments for the first Sat and Sun following surgery, meaning each rat received between 7 and 11 consecutive days of treatment initially. After that, treatments were administered on weekdays across the 12 weeks (Mon – Fri for 5 days / week, but not on Sat and Sun).

This was done for logistic reasons. The long surgeries meant we could only implant 2 per day and we did not want animals implanted on Friday to have treatment started on Day 3. Instead, treated rats implanted on Friday received 7 days of consecutive treatment before a 2 day break for the weekend, whereas animals implanted on Monday received 11 consecutive days of treatment. However, all animals received the same total number of daily treatment sessions at the end of their 12 weeks (62 hours across 84 days).

We have modified the text in this paragraph to make this clearer to the reader.

lines 126-128: "During the first week after surgery, the treated group showed slower recovery compared to the non-treated group, with a significant difference observed on day 7...". How can this phenomenon be explained? In essence, these are the same animals - only trauma, without treatment (as mentioned

Each of the treated rats received 7 days of 1 hour daily electric field treatment prior to this Day 7 BBB score. Please see our response to Reviewer 2 (first of their 'minor comments') for a more detailed reply. In summary, we speculate the observed difference may be related to the daily plugging/unplugging of treated animals, despite efforts to minimize strain. Tethering forces can impact tissue, as demonstrated in brain implants in Vomero et al. (2022). In our case, due to the presence of the acute spinal cord injury, we suspect tethering influenced early inflammation, yet

treated animals still showed better recovery over 12 weeks. Future work aims for a fully integrated system to eliminate this issue. For further clarification we have added this explanation to the discussion.

line 171: Please indicate the area of counting (μm^2 or mm^2) in the spinal cord tissue of Iba-1(+)-cells and GFAP(+)-cells (Fig. 1).

We believe the reviewer is referring to Figure 2. The same dorsal and ventral ROI's shown in Fig. 2C were used for all of the histological analyses in the study (Fig. 2 and 3). This was not clear, so we have added a sentence in the results to indicate this and to clarify that each of these ROIs is 7.1 mm^2 .

line 200: Please indicate the area of counting (μm^2 or mm^2) in the spinal cord tissue of B-Tub(+)-cells, GAP43(+)-cells and 5HT(+)-cells (Fig. 3).

The same dorsal and ventral ROI's shown in Fig. 2C were used for the histological analyses of B-tub(+) and GAP43(+) cells in Fig. 3. We have now made this clear.

lines 306-307: "We were unable to explant any other devices intact at ...". Does this mean that with any removal of the stimulator device (especially when you point out the encapsulation of its body), there is a re-damage to the nervous tissue? Is it possible and if so, how to avoid (or minimize) such re-damage?

The issue was in removing the muscle tissue that was encapsulating the body of the device, above the spinal cord. This required careful dissection with micro-scissors under a microscope, but the polyimide implant was difficult to visualize within the tissue and could be inadvertently severed. In contrast, the subdural portion of the implant slides out relatively easily, and we have not observed any clear evidence of spinal cord re-damage during removal, though confirming this definitively remains challenging.

It's important to note that the device was not originally designed for explantation. However, due to the changes in impedance *in vivo*, we wanted to be able to electrically test the devices afterwards. To address these issues, we are exploring the feasibility of encapsulating the device body within a silicone-based polymer to minimize fibrotic encapsulation and facilitate explantation. Additionally, we are investigating surface coatings for the subdural portion of the implant to reduce tissue adhesion and explore removal safety in future work.

line 513: Please indicate the mass of the animals.

Thank you, this was an oversight. We have now provided information about surgery and final weight ranges of the rats.

line 556: Please describe how the contusion modelling in rats was performed (briefly).

We have added text in the methods to describe how the impact is performed and results in the force over time plots shown in Supplementary Fig. 1

lines 566-567: see lines 120-121. It is unclear, please clarify.

Yes, we agree that the timeline of the treatment is better described in the methods. Therefore, we have revised the first paragraph of results to describe the administration of the electric field

treatment more clearly to the reader.

line 664: What is the idea behind choosing a cryosection thickness of ~1000 μm ?

We prepared 20 μm thick longitudinal sagittal sections from six regions parallel to the midline of the 1 cm tissue blocks and distributed them onto slides. These six regions covered the 1000 μm span shown in Supplementary Fig. 8A. All sections within this span included the lesion, and our a priori expectation was to observe histological differences between treated and non-treated groups, particularly in the tissue directly adjacent to the lesion, between the stimulation site and the injury.

Reviewer #4 (Remarks to the Author):

The authors present a work about the use of low-frequency electrical field to promote axonal regeneration of the lesioned nerve. The injection of long stimulation pulses would benefit from the use of electrodes made of capacitive materials, such as the SIROF. The capability to inject high charge inside the safe voltage limits is critical to minimize tissue damage and facilitate the stability of the electrodes over time. The authors try to validate the stability of these electrodes over time and prolonged stimulation, in both *in vitro* and *in vivo*, which is essential to provide the therapy over weeks.

Another key point of the study is the strategy of removing the dura to reduce the electric field strength and deliver a stronger stimulation while using an order of magnitude less power. This approach has the advantage of reducing the current level, which is critical for the safe injection of long pulses, such as the one required for electric field treatments

In summary, it is recognized the value of exploring the use of high-performing electrodes, in combination with a different implantation strategy, however, the conclusions are not always supported by comprehensive data and statistical significance.

In Figure 5, the authors show the electrochemical characterization of one SIROF electrode, to prove stability after the aging test, and in combination with electrical stimulation. The plots presented show good stability for the impedance and cyclic voltammetry of one electrode, however, statistics on a larger number of electrodes are required to ensure the significance of the results.

We acknowledge the importance of including more electrodes replicates. However, each long-term measurement occupies our stimulator for 90 hours or up to 60 days, making large-scale replication unfeasible with our current resources. The purpose of our paper is also not to present a fully optimized technical system, but to explore this particular application and stimulation domain. The beneficial effects that we thereby are able to report, will in turn motivate further optimization including adhesion layers, and multiplexed stimulation, which will allow the extended stability analysis we understand is requested here. For the present paper our goal was to demonstrate that our solution using SIROF allow us to explore this novel stimulation paradigm which was not possible in the same way using platinum. We use *in vitro* data to support that we achieve sufficient stability to cover the extended stimulation time frame in the *in vivo* experiment, and to compare platinum and SIROF. To comply with the request from the reviewer as far as was possible in the limited time frame, we have included additional data.

We now present triplicates of the 90h continuous stimulation in Fig. 5D-E. CV and EIS of the three specimens are comparable, showing good reproducibility. This was expected as the stimulation charge density of 4 mC/cm^2 is inside the previously found safe injection limit for short μs -pulses and well below the limit for the onset of water electrolysis at the anode for SIROF electrodes during direct current stimulation (More details in Methods under *Electric field treatment sessions*). For the long-

term study, we now added the replicate we initially conducted in Fig. 5G-H. Additionally, we replicated the experiment mimicking the *in vivo* treatment. This time, instead of shorting four electrodes as CEs and stimulating two WEs consecutively, we used two electrodes as CEs and one as a WE. This adjustment was based on our hypothesis at the time that consecutive stimulation contributed to layer deposition due to polarization. This 1-hour daily stimulation for 60 days was conducted at 37°C in 1×PBS, and, consistent with previous long-term experiments, the electrodes remained functional.

Interestingly, we observed that the deposited layer previously reported on CEs now formed on the WE. These findings are shown in Fig. 5G-H and Supplementary Fig. 4E-F. Chemical analysis revealed that the deposited layer primarily consists of C=C and/or C-H (Supplementary Fig. 4G). Since no other carbon sources are present in the benchtop experiment, we propose that dissolved CO₂ acts as the carbon source in this reaction. Notably, similar carbon deposition was not observed during 90h of continuous stimulation, suggesting that the reaction is driven not by stimulation itself but rather by polarization after stimulation. This is supported by the carbon reduction peak in CVs at -400 mV vs Ag/AgCl, which aligns with the interpulse voltage observed in voltage transient measurements (Fig. 5C).

While this low-voltage, slow-rate carbon electrodeposition on SIROF is a noteworthy phenomenon, relevant in all studies using SIROF electrodes, it is beyond the scope of this work especially considering that this was not seen on implanted electrodes. Our primary aim was to assess electrode stability in benchtop experiments over the treatment period. To this end, we have demonstrated stability across multiple conditions (90h continuous stimulation, 60 days at 37°C, and 18 days at 55°C).

In summary, to address the reviewers' request for significance, we extended the 90 hours continuous stimulation to triplicates, and we have provided an additional long-term stimulation data point, which required substantial time to replicate. Importantly, despite carbon deposition, the electrodes remained fully functional, as confirmed by CV and EIS. Thus, our data further supports the conclusion that electrodes remain stable in benchtop experiments. We have summarized the above findings in a new written Supplementary Note 1.

It remains an open question whether similar carbon deposition occurs *in vivo*. In this study, high-resolution images did not reveal deposited carbon in tissue (Supplementary Fig. 7). However, further investigation is warranted, which we plan to address in a separate study. We have already initiated experiments to determine time points when carbon deposition becomes detectable in CVs.

In the paper, the use of capacitive SIROF electrodes is presented as an advantage to inject long pulses of 250 ms in the safe voltage window. The electrochemical characterization should include the measure of the voltage polarization of the stimulation pulses used in this study to evaluate if the voltage shift at the electrode-tissue interface is inside the voltage-safe limits.

Voltage transients are within the safe voltage window and now presented in Fig. 5C.

In Figure 6a, the impedance spectrum of the electrodes *in vivo* is used to ensure the functionality of the electrodes over time. The fact that the magnitude is below the open circuit potential limit, is it enough to claim that the electrodes are functional?

Characteristics such as the shape of the impedance spectrum need to be considered. The presence of a 'noisy measurement' for the impedance magnitude of some electrodes, as well as the change of the phase, are indications that the electrodes cannot function properly, at least in the *in vivo* settings. How is the measurement of the cyclic voltammetry over weeks of implantation?

We agree with the reviewer that impedance magnitude being below the OCP limit alone is insufficient to conclusively determine electrode functionality. Accordingly, in our discussion, we state: "*These results suggest that the electrodes effectively delivered stimulation over the treatment period, with*

some individual variation where it is not possible to fully exclude earlier interruption due to delamination”.

Our conclusion is based on multiple considerations beyond impedance measurements. Notably, only one out of ten stimulation electrodes exhibited substantial delamination that would render it non-functional (Fig. 7B). Additionally, implants explanted after three weeks remained functional following tissue removal, despite initially exhibiting high impedance (Supplementary Fig. 10). However, the most compelling evidence supporting effective stimulation delivery is the observed functional improvement in treated animals.

To enhance future assessments of implant functionality, we propose methodological refinements, as stated in our discussion: *“Future studies should incorporate additional methods (e.g., monitoring the current or testing the threshold to evoke motor function) to assess implant functionality in situ or utilize coatings that reduce encapsulation, thereby facilitating explantation.”* CV was not performed in this study.

Since the *in vivo* functionality of the electrodes *in vivo*, is intended in the perspective of delivering the therapeutic treatment over time, the voltage polarization of the stimulation pulses should be included in the characterization. After the increase of the impedance over weeks of implantation, can the electrodes still inject the pulses inside the safe voltage limits to avoid faradaic reactions that can damage both electrodes and tissue? Can the voltage compliance of the stimulator, deliver the voltage needed to stimulate even after several weeks?

Voltage transients are now presented in Fig. 5C. Between the treatment sessions, there is a minimum 24 hour no-stimulation period, during which we expect the electrodes to resolve polarization. The MultiChannelSystem stimulator used in this study did not allow for voltage transient recording.

We observed no indications of excessive tissue reactions due to stimulation, which would be expected if charge delivery were dominated by faradaic reactions.

Since EIS applies only small voltage signals, the recorded impedance does not directly reflect the voltage excursion during current pulsing. As a result, the exact *in vivo* voltage excursion remains unknown. However, the low voltage transients observed *in vitro* (Fig. 5C), the stimulator’s high voltage compliance (16V), and the functional improvements observed in treated animals collectively suggest that stimulation was successfully delivered.

The presence of tissue encapsulation around the device, produced by the foreign body reaction (FBR) after weeks of implantation, raises a question in relation to the advantage of removing the dura when the treatment is extended over time. to improve the penetration of the electric field and the efficacy of the treatment. In the text it is cited a study (doi: 10.1063/5.0163264) to support the statement that the presence of FBR, has little effect on the field distribution in the white and gray matter. However, in this model, the FBR is considered in a device implanted on top of the dura. In the case of subdural electrodes, how is it expected to change the electric field after weeks of implantation? The advantage of having electrodes below the dura to improve the penetration of the electric field and the efficacy of the treatment is still an advantage after weeks?

In the cited study, the FBR was modelled by reducing the conductivity of a 0.2 mm tissue layer directly beneath the electrodes. The findings indicated that while the conductivity of the FBR-affected tissue influenced the EF strength at the electrode-tissue interface, it had minimal impact on EF within the spinal cord. We expect similar results for subdural electrodes, as a thin layer of FBR-tissue with decreased conductivity will not deprive much power from the EF and thus will have little effect. This effect is general and, therefore, independent of electrode placement.

The second question posed by the reviewer concerns the comparative advantages of subdural versus epidural electrode placement, given that subdural implantation is more invasive and may lead to an increased FBR. Our work represents a step forward in addressing this question. To date, the field has primarily focused on functional epidural stimulation, despite its inherent limitations, such as reduced precision due to the shunting effect of CSF and increased power consumption. This preference is largely driven by the advantage of a less invasive surgical procedure. Nevertheless, research on subdural stimulation also exists. Notably, intraspinal microstimulation, which involves implanting thin wires into the lumbar enlargement of the spinal cord, has demonstrated superior functional outcomes compared to subdural stimulation, particularly in terms of generating movement with reduced fatigue in animal models (Saigal et al. 2004). In our study, we demonstrate that subdural electrode placement is feasible and leads to functional improvements. However, to fully address the reviewer's question, further research is required to comprehensively evaluate the potential and limitations of both implantation strategies. We now acknowledge that more work is needed to assess which solution is better for translation to patients in the discussion (l. 370).

Reference: Saigal R, Renzi C, Mushahwar VK. (2004). Intraspinal microstimulation generates functional movements after spinal-cord injury. *IEEE Transactions on Neural Systems and Rehabilitation Engineering*, 12(4):430-40. <https://doi.org/10.1109/tnsre.2004.837754>

Has been any test or analysis performed to evaluate that not only the electrodes but also the polyamide encapsulation of the device is stable over 12 weeks of in vivo implantation, to exclude liquid permeation inside the polymeric sandwich which can produce the electrodes short-cut and consequent change in the stimulation current spreading?

Comparable polyimide implants from our group have been stable over similar or longer time periods: In Ref. 47 for 12 weeks in rats, in Ref. 48 for over 5 months in mice, in Ref. 49 for 12 weeks in rats. In none of these studies failure of the insulation has been observed. We purposely only cited studies in the manuscript from our group, as variations in fabrication may influence the implant's stability. In our study, liquid permeation was not observed during benchtop long-term experiments. To further address the reviewer's comment we have performed FIB-SEM on three randomly chosen implants from the treated group on different areas to evaluate the status of the connection lines. As expected, the connection lines and the insulating PI in all investigated specimens are intact. We have added this data to a new Supplementary Fig. 13 and added it to the discussion (l. 457). In the future we envision a device which is only implanted for a limited time to enable the axonal regeneration after SCI.

The authors use 4 electrodes shorted as a counter path for the stimulation current. It is visible in the design a big rectangular electrode, which I suppose was designed in the first place, to work as a counter. Could the authors comment on the choice of using 4 electrodes shorted, instead of the bigger rectangular electrode?

The large rectangular electrodes at the tip were not originally designed to function as a counter electrode in stimulation. Utilizing the large electrodes at the tips as counter electrodes during stimulation would increase the volume of stimulated tissue, thereby reducing the EF strength. We aimed to focus the EF into the lesion utilizing the dedicated stimulation electrodes. Shorting four stimulation electrodes and using them as counter electrodes was necessary because the stimulator permits only a single ground pathway. We have included the explanation for shorting four stimulation electrodes in the method section.